



**A versatile, refrigerant-free cryofocusing-thermodesorption**
**unit for preconcentration of traces gases in air**
F. Obersteiner[1], H. Bönisch[2], T. Keber[1], S. O'Doherty[3] and A. Engel[1]
[1] Institute for Atmospheric and Environmental Science, Goethe University Frankfurt,
Frankfurt, Germany
[2] Institute of Meteorology and Climate Research, KIT, Karlsruhe, Germany
[3] School of Chemistry, University of Bristol, Bristol, United Kingdom
*Correspondence to*: F. Obersteiner, obersteiner@iau.uni-frankfurt.de
**Abstract.** We present a compact and versatile cryofocusing-thermodesorption unit, which we
developed for quantitative analysis of halogenated trace gases in ambient air. Possible appli-
cations include aircraft-based in-situ measurements, in-situ monitoring and laboratory opera-
tion for the preconcentration of analytes from flask samples. Analytes are trapped on adsorp-
tive material cooled by a Stirling cooler to low temperatures (e.g. −80 °C) and desorbed sub-
sequently by rapid heating of the adsorptive material (e.g. +200 °C). The setup neither in-
volves exchange of adsorption tubes nor any further condensation or refocusation steps. No
moving parts are used that would require vacuum insulation. This allows a simple and robust
single-stage design. Reliable operation is ensured by the Stirling cooler, which does not re-
quire refilling of a liquid refrigerant while allowing significantly lower adsorption tempera-
tures compared to commonly used Peltier elements. We use gas chromatography - mass spec-
trometry for separation and detection of the preconcentrated analytes after splitless injection.
A substance boiling point range of approximately −80 °C to +150 °C and a substance mixing
ratio range of less than 1 ppt (pmol mol$^{-1}$) to more than 500 ppt in preconcentrated sample
volumes of 0.1 to 10 L of ambient air is covered, depending on the application and its analyti-
cal demands. We present the instrumental design of the preconcentration unit and demonstrate
capabilities and performance through the examination of injection quality, analyte break-
through and analyte residues in blank tests. Application examples are given by the analysis of
flask samples collected at Mace Head Atmospheric Research Station in Ireland using our la-
boratory GC-TOFMS instrument and by data obtained during a research flight with our in-situ
aircraft instrument GhOST-MS.





# 1  Introduction

Atmospheric trace gases introduced to or elevated in concentration in the environment by human activities often show adverse environmental impacts. Prominent examples are chlorofluorocarbons (CFCs) and their intermediate replacements, hydrochlorofluorocarbons (HCFCs), which deplete stratospheric ozone (Farman et al., 1985; Molina and Rowland, 1974; Montzka et al., 2011; Solomon, 1999). Present-day CFC-replacements, namely hydrofluorocarbons (HFCs), have zero ozone depletion potentials (ODPs) but are still potent greenhouse gases like CFCs and HCFCs (Hodnebrog et al., 2013; Velders et al., 2009). Another example are non-methane hydrocarbons (NMHCs), which produce harmful tropospheric ozone in the presence of nitrogen oxides (Haagen-Smit and Fox, 1956; Marenco et al., 1994; Monks et al., 2015).

Many of the species found in the compound classes named above show atmospheric concentrations too low for direct detection and quantification by means of instrumental analytics. Therefore, a preconcentration step is required. The method of cryofocusing-thermodesorption is a common technique for that purpose (e.g. Aragón et al., 2000; Demeestere et al., 2007; Dettmer and Engewald, 2003; Eyer et al., 2016; Hou et al., 2006). In principal, an ambient air sample from either a sample flask or continuous flow for online measurement is preconcentrated on adsorptive material at a specific adsorption temperature, $T_A$. If $T_A$ is significantly below ambient temperature, this step is referred to as "cryofocusing" or "cryotrapping". Trapped analytes are re-mobilized subsequently by heating the adsorptive material to a desorption temperature $T_D$ and flushed e.g. onto a gas chromatographic column with a carrier gas and detected with a suitable detector.

The primary motivation for the development of the instrumentation described in this manuscript was halocarbon analysis in ambient air. More specifically, there were no commercial instruments available which met the requirements of remote in-situ and aircraft operation: compact (as small as possible), lightweight (<5 kg), safe containment of working fluids and preferentially cryogen-free, pure electrical operation. Liquid cooling agents like liquid nitrogen ($LN_2$) or argon (LAr) (e.g. Apel et al., 2003; Farwell et al., 1979; Helmig and Greenberg, 1994) offer large cooling capacity but are difficult to operate on board of an aircraft due to safety restrictions and supply demand, e.g. when operating the aircraft from remote airports. Compression coolers (e.g. Miller et al., 2008; O'Doherty et al., 1993; Saito et al., 2010) offer less cooling capacity in terms of heat lift compared to liquid cooling agents and are relatively



large in size and weight compared to widespread Peltier type cooling options (Peltier ele-
ments; e.g. de Blas et al., 2011; Simmonds et al., 1995; commercial thermodesorbers available
from e.g. Markes or PerkinElmer). Peltier elements have the advantage of being very small
and requiring only electrical power for cooling. However, their cooling capacity and mini-
mum temperature cannot compete with compression- and refrigerant-based coolers. Stirling
coolers pose an in-between solution, well-suited for maintenance-free remote operation: like
Peltier coolers, they only require electrical power, do not contain any potentially dangerous
working fluids (only helium) or cryogens but have a significantly higher cooling capacity.
While not being as powerful as refrigerant-based coolers (LN$_2$, LAr), they still have compara-
ble minimum temperatures. To our knowledge, the use of Stirling coolers for similar purposes
like the one described here is rare with few published exceptions like the preconcentration of
methane by Eyer et al. (2016) or the trapping of CO$_2$ as a carbon capture technology by Song
et al. (2012).
The principal design of the cryofocusing-thermodesorption unit in description was developed
for the airborne in-situ instrument GhOST-MS (Gas chromatograph for the Observation of
Tracers – coupled with a Mass Spectrometer, Sala et al., 2014) and successfully used during
three research campaigns up to now – 2011: SHIVA (carrier aircraft: DLR FALCON), 2013:
TACTS (carrier aircraft: DLR HALO), 2015/2016: PGS (carrier aircraft: DLR HALO). To
extend the substance range, we then developed similar cryofocusing-thermodesorption units
for our other GC-MS instruments (Hoker et al., 2015; Obersteiner et al., 2016), which are
currently operated in the laboratory. Both detailed description and characterisation of the pre-
concentration unit were not discussed in the publications Hoker et al. (2015), Obersteiner et
al. (2016) (laboratory setups) and Sala et al. (2014) (aircraft instrument). Within this manu-
script, a general instrumental description is given in section 2, which is applicable for all the
named setups. Characterisation results discussed in section 3 are based on the latest version of
the laboratory setup (Obersteiner et al., 2016). To demonstrate the versatility and reliability of
the setup, application examples are given in section 4 for sample analysis in the laboratory as
well as in-situ aircraft operation. Results are summarized and conclusions are drawn in sec-
tion 5.



## 2   Instrumentation

This section gives a description of principal components of the sample preconcentration unit and is valid for all our analytical setups presented in Sala et al. (2014), Hoker et al. (2015) and Obersteiner et al. (2016). The following section 2.1 outlines the general measurement procedure and gas flow as well as its integration into a chromatographic detection system. Sections 2.2 and 2.3 describe the implementation of the main operations of the unit; cooling ("trapping", i.e. preconcentration of analytes) and heating (desorption of analytes). A preconcentration system can always only be as good as the analytical set-up behind it. The preconcentration system described here has been designed for the coupling with a chromatographic system but in principle could also be adapted for coupling with other techniques. Specific technical components of the instrumentation used in this work to characterise the preconcentration unit will be listed in section 3.

### 2.1   Measurement procedure and gas flow in GC application

For the preconcentration of analytes, the sample is flushed through a micro packed column of cooled adsorptive material. Analytes are "trapped" on the adsorptive material as the steady state of adsorption and desorption is strongly shifted towards adsorption by the low temperature of the adsorptive material. By subsequent rapid heating of the adsorptive material, the steady state is instantaneously shifted towards desorption ("thermodesorption"). Formerly trapped analytes are flushed backwards onto the warm chromatographic column with a carrier gas. There is no further refocusing or separation step, except for higher-boiling compounds on the GC column itself. **Figure 1** shows a flow scheme of the setup. The outflow of the sample loop during preconcentration ("stripped air"; mainly nitrogen and oxygen) is collected in a previously evacuated reference volume for analyte quantification (2 L electro-polished stainless steel flask; volume determination by pressure difference). A mass flow controller (MFC) is mounted between sample loop and reference volume for sample flow control. The MFC can also be used for sample volume determination e.g. for sample volumes larger than the reference volume. Hardware control is implemented with a LabVIEW cRIO assembly (compact, reconfigurable input output; National Instruments Inc., USA) using self-written control software. It operates the preconcentration unit automatically, i.e. controls system parameters like sample loop temperature by cooling and heating concomitant with system states like preconcentration, desorption etc.





## 2.2 Cryofocusing: sample loop and cooling technique

A stainless steel tube with 1/16" outer diameter (OD) and 1 mm inner diameter (ID) is used as sample loop. The tube is packed with adsorptive material and placed inside an aluminium cuboid ("coldhead") which is cooled continuously to maintain a specific adsorption temperature. **Figure 2** shows a technical drawing of sample loop and coldhead. The coldhead can contain two sample loops; in this case one of them is an empty stainless steel tube with 1/16 inch OD and 1 mm ID to characterize the sample loop heater. For that purpose, a thin temperature sensor is inserted into the empty tube. To save space and avoid mechanical, moving parts, the sample loop is not removed from the coldhead during desorption. It is insulated and thereby isolated electrically by two layers of glass silk and four layers of Teflon shrinking hose. The insulation is a variable parameter which determines the rate at which heat is exchanged between sample loop and coldhead. Consequently, it determines coldhead warm-up rate during desorption and sample loop cool-down rate after desorption. More insulation would result in longer cool-down time after desorption but also to less heat flowing into the cold head, thus to lower possible temperature of the cold head. The insulation used represents a compromise that works well for the application presented here but could potentially be improved by e.g. using a ceramic insulator. The coldhead itself is insulated towards surrounding air with 45 mm of Aeroflex HF material (Aeroflex Europe GmbH, Germany).

The Stirling cooler used for cooling offers the advantage of requiring only electrical power while providing a relatively large cooling capacity at very low minimum temperatures. The latter are comparable to liquid nitrogen in case of Sunpower CryoTel MT, CT and GT Stirling coolers, with maximum heat lifts of 5 W to 16 W at −196 °C according to the manufacturer. Heat that is removed from the coldhead by the Stirling cooler has to be released to the surrounding air; either directly by an air-fin heat rejection or indirectly by a water coolant system mounted to the cooler's warm side. The cooler should maintain a defined adsorption temperature $T_A$ of the sample loop over the series of measurements. However, during thermodesorption, a certain amount of heat is transferred to the coldhead as the sample loop is kept directly inside with only a small amount of insulation. Excess heat has to be removed by the Stirling cooler to regain $T_A$ for the preconcentration of the next sample. The preconcentration unit is attached to a gas chromatograph; therefore, the gas chromatographic runtime allows coldhead and sample loop to cool down after thermodesorption and return to $T_A$ before preconcentrating the next sample.





Besides chromatographic runtime, various factors determine the minimum cycle time (i.e.
sample measurement frequency) including:

- targeted adsorption temperature $T_A$
- Stirling cooler's cooling capacity (i.e. heat lift around $T_A$) and coldhead insulation as well as ambient temperature
- thermodesorption duration and $T_D$ as well as insulation of the sample loop
- volume of the sample to preconcentrate and preconcentration flow

To give a practical example, **Table 1** shows cycle times derived from routine operation data.
With the laboratory setup, a total time per measurement of 18.6 minutes is necessary if
$T_A = -120\,°C$ and $T_D \approx 200\,°C$ is desired – mainly determined by the time needed to compen-
sate the warm-up of the coldhead during desorption. This minimum time interval significantly
shortens to 8.5 minutes if $T_A$ is increased to $-80\,°C$ (same $T_D$). Data from the in-situ setup
shown in **Table 1** demonstrates that even shorter cycle times of 4.1 minutes are possible with
a decreased preconcentration volume (100 mL instead of 500 mL; requiring a detector that is
sensitive enough) and a slightly higher $T_A$. General measures to increase the number of meas-
urements per time would be to increase the preconcentration flow, reduce the sample size (see
in-situ setup), improve the coldhead and sample loop insulation and increase the cooling ca-
pacity.
After desorption, sample loop temperature drops in an exponential decay shaped curve due to
the decreasing temperature difference between coldhead and sample loop. After a desorption
at $T_D \approx 200\,°C$, sample loop and coldhead temperature reached similar temperatures after ap-
proximately 30 s cool-down time ($T_A = -80\,°C$). The cool-down time increases to about 90 s
at $-120\,°C$ cold head temperature. Considering the total run times shown in (**Table 1**), sample
loop cool-down time is not a limiting factor to the overall cycle time. Consequently, thermal
insulation of the sample loop could still be increased, thereby decreasing coldhead warm-up
during desorption.



## 2.3 Thermodesorption: sample loop heater
Depending on the targeted substance class to analyse and the analytical technique, the re-
quirements for thermodesorption will differ. In case of a gas chromatographic system for
analysis of volatile compounds, these requirements are:
- a fast initial increase in temperature to yield a sharp injection of highly volatile
analytes onto the GC column,
- no overshooting of a maximum temperature in case of thermally unstable sample
compounds or adsorptive material (e.g. HayeSep D, $T_D < 290$ °C)
- preservation of the desorption temperature over a time period for desorption of
analytes with higher boiling points
- good overall repeatability, especially of the injection of highly volatile analytes
Desorption heating is implemented by pulsing a direct current (max. 12 V / 40 A, relay:
Celduk Okpac; spec. switching frequency 1 kHz, Celduk Relays, France) directly through the
sample loop tubing which has a resistance of ~0.5 Ω. A temperature sensor (Pt100, 1.5 mm
OD) was welded to the outside of the sample loop tubing (see also **Figure 2**), for feedback
control of the heater temperature. However, mainly due to the thermal mass of the sensor and
its proximity to the coldhead (despite the insulation), it was found to give no representative
values for temperature inside the sample loop during desorption. Differences of around
100 °C were found in comparison to temperature measured within the sample loop (equilibri-
um state; after 2-3 minutes of continuous heating). Nevertheless, the temperature sensor can
be (after being characterised) used for feedback control as the indicated values are reproduci-
ble. As an alternative to feedback control, a deterministic heater with prescribed output set-
tings can be used. For security reason, measured coldhead and sample loop temperature have
to be used as heater shutdown triggers in this case.
**Figure 3** shows a comparison of temperature sensor data from in- and outside the empty sam-
ple loop as well as the coldhead. Very good results were achieved with a two-stage, determin-
istic heater setup with a fast heat-up, a small overshoot between stage 1 and 2 of the heating
phase and preservation of $T_D$ with only a small drift and fluctuation. With the described heater
setup, $T_D$ can be reached within a very short time of approximately 3 seconds. Initial heating
rates (first second of heat pulse) were calculated to be more than 200 °C s$^{-1}$ depending on the
power output setting. As the sample loop is getting warmer, heating rate drops resulting in a
mean heating rate of about 80 °C s$^{-1}$ during stage 1.
If a deterministic heater is used instead of a feedback controlled heater, sample loop tempera-
ture becomes directly dependent on coldhead temperature (more precisely: heat flow from the
sample loop into the coldhead). Consequently, higher output settings are necessary at lower
coldhead temperatures to achieve comparable temperatures. On the other hand, if the cold-
head gets warmer, sample loop temperature increases as well. This effect can be observed in
**Figure 3** as a slight upward drift of the sample loop temperature (red curve, temperature
measured within the sample loop) during stage 2. The absolute temperature differences caused
by this drift as well as the oscillation amplitude are small (approximately 20 °C min. to max.
and 4 °C standard deviation without trend correction) compared to the temperature difference
between coldhead and sample loop during heating (about 300 °C).
Besides the problem of differing inner and outer temperature of the sample loop during heat-
ing, temperature was not found to be distributed homogeneously alongside the empty sample
loop inside the coldhead. Temperature differences of up to ±30 °C at 200 °C mean tempera-
ture were observed with the current setup if measuring temperature at different points within
the sample loop, potentially due to (a) difficulties in accurately measuring the inner tempera-
ture (wall contact of sensor) and (b) inhomogeneity in sample loop insulation as well as varia-
tions in tubing wall width or carbon content leading to an inhomogeneous electrical resistance
and thus an inhomogeneous distribution of heat. These temperature variations might be differ-
ent or ideally negligible in the sample loop packed with adsorptive material. However, the
finding underlines the importance of an insulation as homogeneous as possible and suggests
that "cold points" (possibility of insufficient desorption) as well as "hot points" (possibility of
adsorptive material or analyte decomposition) are possible along the sample loop, which has
to be taken into consideration when setting up and testing the preconcentration setup, i.e. to
not exceed the temperature limit of the adsorptive material.





## 3 Characterisation

This section discusses characterisation results (section 3.2 and 3.3) obtained with the GC-TOFMS instrument described in Obersteiner et al. (2016) as it covers the widest substances range (see supplementary information) and therefore allows the most differentiated analysis. A brief description of this analytical instrument is given in the following section 3.1; see Obersteiner et al. (2016) for details on GC and MS. We consider these results to be valid in principle also for our other GC-MS setup discussed by Hoker et al. (2015) and the GhOST-MS described by Sala et al. (2014) as all preconcentration setups rely on the same principal setup and similar components are used.

### 3.1 Analytical instrument

A Sunpower CryoTel CT free piston Stirling cooler (Ametek Inc., USA) is used for cooling of the coldhead. In the described setup, a water coolant system (Alphacool, Germany) originally intended for cooling of a personal computer's processing units removes heat from the Stirling cooler's heat rejection. Sunpower Stirling coolers are optionally also available with an air-fin heat rejection that requires a continuous air stream during operation. For sample loop heater control, a pulse-width modulation (PWM; 20 ms period, 1 µs minimum width) with a prescribed output is used (deterministic heater; see section 2.3). Heater operation during desorption is separated into a short initial "heat-up" stage with a high output of the PWM and a longer "hold" stage with lower heater output to maintain desorption temperature. The sample loop is packed with adsorptive material over a length of approximately 100 mm (~20 mg). Two different adsorptive materials were used in different sample loops installed in the course of this work; HayeSep D, 80/100 mesh (VICI International AG, Switzerland) and Unibeads 1S, 60/80 mesh (Grace, USA).

A Bronkhorst EL-FLOW F-201CM (Bronkhorst, the Netherlands) is used for sample flow control (downstream of the sample loop in order to avoid contamination) in combination with a Baratron 626 pressure sensor (0-1000 mbar, accuracy incl. non-linearity 0.25 % of reading, MKS Instruments, Germany) for analyte quantification by pressure difference measurement. An Agilent 7890 B gas chromatograph (GC) with a GS GasPro PLOT column (Agilent Technologies, Inc. USA; 0.32 mm inner diameter) using a ramped temperature program (45 °C to 200 °C with 25 °C min$^{-1}$) and backflush option is used for analyte separation. Purified helium 6.0 is used as carrier gas (Praxair Technologies Inc., German supplier; purification system:



Vici Valco HP2). For analyte detection, a Tofwerk EI-TOF (model EI-003, Tofwerk AG,
Switzerland) mass spectrometer (MS) is attached to the GC. All samples are dried using mag-
nesium perchlorate kept at 80 °C prior to preconcentration. Artificial additions of analytes to
the sample from the dryer were excluded by comparing measurements of dried and undried
blank gas. All tubing upstream of the sample loop was heated to >100 °C to avoid substance
loss to tubing walls.
**Figure 4** shows a typical chromatogram from an ambient air sample for three selected
mass-to-charge ratios (m/Q). Two different adsorptive materials were used in the course of
this work (HayeSep D, Unibeads 1S) which showed partly differing adsorption and desorption
properties; results are discussed separately if appropriate. To achieve high measurement pre-
cision and minimum uncertainties introduced by the preconcentration unit, both the analyte
adsorption (preconcentration) and analyte desorption (injection) into the chromatographic
system have to be quantitative and repeatable. The following section describes tests and re-
sults for the characterisation of both aspects.

## 3.2  Adsorption

The sample loop essentially is a micro packed chromatographic column with a limited surface
area where sorption can take place. The low temperature during sample preconcentration
shifts the steady state of analyte partitioning between mobile and solid phase mostly to the
solid phase. This preconcentration technique "strips" the air of its most abundant constituents;
nitrogen, oxygen and argon. Other, less volatile but still very abundant constituents like $CO_2$
are however trapped, depending on adsorption temperature. Elution of such species from the
GC column after thermodesorption and injection can cause problems with regard to chroma-
tography as well as detection, depending on GC configuration and detection technique. With
the setup described here, the elution of $CO_2$ limits the analysable substance range as the detec-
tor shows saturation during the elution of $CO_2$. Regarding preconcentration of targeted ana-
lytes, the concept of an adsorption-desorption steady state suggests that at a certain point a
breakthrough of analytes occurs, depending on a combination of loading of the solid phase
with sample molecules and time to achieve steady state, in turn influenced by sample flow
rate and pressure. Consequently, the maximum possible sample volume and/or minimum du-
ration of preconcentration are dependent on the adsorptive material used, volatility (and con-
centration) of the targeted analytes as well as sample flow rate and pressure. For typical sam-
ple volumes of 0.5 L and 1.0 L (at standard temperature and pressure) and a constant sample




back pressure of 2.5 bar abs., no significant impact of sample preconcentration flow was
found within the tested range of 50 mL·min$^{-1}$ to 150 mL·min$^{-1}$ for any of the analysed sub-
stances. Higher or lower flow rates and pressure were not possible or suitable for practical
reasons like flow restriction and valve operating pressure.
Substance breakthrough (i.e. substance-specific adsorption capacity) was analysed in volume
variation experiments, comprising measurements of the same reference air with preconcentra-
tion volumes of up to 10 L and referencing the volume-corrected detector response against
default preconcentration volumes of e.g. 1 L ("relative response"). Quantitative trapping is
then indicated by a relative response of 1; a relative response <1 would indicate an underesti-
mation (i.e. loss by breakthrough), a relative response of >1 would indicate an overestimation
(i.e. increase by a memory effect from the preceding sample). To structure the following dis-
cussion, two classes of substances are formed and treated separately: "medium volatile sub-
stances" with boiling points > −30 °C (e.g. CFC-12, $CCl_2F_2$) and "highly volatile substances"
with boiling points < −30 °C (e.g. HFC-23, $CHF_3$). The substances discussed are selected
based on the criteria volatility and (preferably high) concentration. The adsorption of sub-
stances with lower volatility (BP > 30 °C) was assumed to be quantitative. Results discussed
in the following are displayed in **Table 2**.
***Medium volatile substances.*** As a reference for halocarbon analysis, CFC-12 ($CCl_2F_2$) and
CFC-11 ($CCl_3F$) were chosen due to their high mixing ratios of about 525 and
235 pmol·mol$^{-1}$ (ppt, parts per trillion) in present-day, ambient air and moderate volatility
with boiling points of −29.8 °C and +23.8 °C. For a volume of 10 L preconcentrated air on the
Unibeads 1S sample loop, both substances showed a deviation from linear response of
+0.6 % ± 0.42 % for CFC-12 and +0.6 % ± 0.22 % respectively for CFC-11. The positive
deviation from linearity is still found within the 3-fold measurement precision determined for
the experiment and could potentially be an artefact of the detector used which tends to slightly
overestimate strong signals and underestimate weak signals; see section 3.4 in
Obersteiner et al. (2016). Hence, no significant breakthrough or detector saturation was ob-
served for both substances CFC-12 and CFC-11.
***Highly volatile substances.*** More volatile compared to CFC-12 and CFC-11 but similar in
mixing ratio is carbonyl sulfide (COS) with a boiling point of −50.2 °C and an ambient air
mixing ratio of typically around 500 ppt. Against 1 L reference sample volume (sample
mixing ratio: 525 ppt), COS showed a quantitative adsorption up to 5 L on the Unibeads 1S



sample loop with a deviation from linear response of +0.9 % ± 0.80 %. At 10 L sample
volume, a breakthrough occurred giving a deviation from linear response of
−35.2 % ± 0.52 %. The substance analysed with highest volatility was HFC-23 with a boiling
point of −82.1 °C and a current background air mixing ratio of ~40 ppt. Referenced against a
sample volume of 0.5 L, significant breakthrough occurred at a sample volume of 2.5 L with a
deviation from linear response of −39.2 % ± 2.75 %. The highest sample volume quantitative-
ly adsorbed in the experiment was 1.0 L with a relative response of −0.3 % ± 2.75 %
(HayeSep D sample loop). A similar behaviour was observed for ethyne ($C_2H_2$), with a subli-
mation point of −80.2 °C, a mixing ratio of approximately 610 ppt in the sample and a devia-
tion from linear response of −20.2 % ± 1.22 % at 2.5 L sample volume (HayeSep D sample
loop). However, ethyne was also analysed on the Unibeads 1S sample loop which gave a quite
different result with a deviation from linear response of +10.1 % ± 0.51 %, thus breakthrough
did not occur. The positive, non-linear response is caused potentially by a system blank (see
also section 3.3). Unfortunately, HFC-23 could not be analysed in ambient air samples for
comparison on the Unibeads 1S sample loop as its ion signals are masked by large amounts of
$CO_2$ still eluting from the GC column at the retention time of HFC-23.
Concluding, the adsorption process was found to be substance specific as both HFC-23 and
ethyne are comparably volatile but significantly less ethyne broke through despite its 15-fold
elevated mixing ratio (Unibeads 1S sample loop). The comparison of ethyne breakthrough on
the HayeSep D and Unibeads 1S sample loop suggests that the adsorption process is depend-
ent on the chosen adsorptive material. A comparison of adsorptive materials is however not
the focus of this work; such a comparative adsorption study was e.g. conducted for methane
($CH_4$) preconcentration by Eyer et al. (2014). From the comparison of the breakthrough ob-
served for COS and the quantitative adsorption of CFC-12 and CFC-11, it can be concluded
that volatility is the primary factor that determines breakthrough. Quantitative adsorption is
not limited by principal adsorption capacity (i.e. the absolute number of molecules adsorbed)
of the adsorptive material and material amount for a sample volume of up to 10 L and an ad-
sorption temperature of −80 °C.



### 1  3.3  Desorption

While adsorption is characterised by the quantitative trapping of highly volatile substances,
desorption is characterised by sharpness and repeatability of the injection represented by
chromatographic peak shape and retention time variance (qualitative aspect; section 3.3.1) as
well as the amount of blank residues (quantitative aspect; section 3.3.2). Blank residues
("memory effect") have to be divided into residues that remain on the adsorptive material
after desorption ("preconcentration residues" or "preconcentration memory effect") and resi-
dues that remain in the analytical setup (tubing etc.) upstream of the sample loop, thus had not
reached the sample loop ("system residues" or "system memory effect").

### 10  *3.3.1  Peak shape and retention time stability*

To demonstrate injection sharpness, **Figure 5 A** shows the chromatographic signal of CFC-11
eluted from the GC column kept isothermal at 150 °C and **Figure 5 B** the chromatographic
signal as observed with the ramped GC program. Both signals generally show a Gaussian
peak shape with a slight tailing of the right flank. In comparison, the "unfocused" signal from
the isothermal column reflecting the sharpness of the direct injection is wider by a factor of
~3 but still narrow enough to allow for good peak separation in most standard GC methods
with runtimes between 10 to 30 minutes; the full peak width at half maximum (FWHM) was
calculated to be 6.3 s (0.10 min) for the isothermal peak and 2.0 s (0.03 min) for the focused
peak.
Injection quality can further be judged by the stability of retention times of the first chromato-
graphic signals obtained with the ramped GC program, as these are only very little influenced
by the chromatographic system (in particular there is nearly no refocusing on the chromato-
graphic column). **Table 3** shows retention times and their variability expressed as relative
standard deviation and variance as well as the chromatographic signal width (FWHM) of the
respective substance. Variances are less than 0.02 s on average. Together with signal width,
they decrease reversely proportional to retention time, which shows the increasing influence
of chromatographic separation (from HFC-23 to CFC-11 in **Table 3**). Even at incomplete re-
focusation by gas chromatography, the desorption procedure of the preconcentration unit
gives close to Gaussian peak shapes except a slight tailing of the right flank. The tailing effect
could potentially be reduced by refocusing the high-volatile analyte fraction on a second sam-
ple loop. The high repeatability of the injection is shown by the low variability in retention
time of the first signals in the chromatogram (**Table 3**).



### 3.3.2 Analyte residues

Analyte residues can originate from inherent system *contamination* or constitute a remainder from the previous sample (*memory effect*). Analyte residues were investigated with (a) an unloaded injection after multiple 1 L ambient air sample injections, i.e. subsequent thermode-sorption of the sample loop without switching to load-position between runs (see **Figure 1**) and (b) the preconcentration of 1 L helium from the carrier gas supply using the same path as the sample, including dryer etc. after multiple 1 L ambient air sample measurements. Analyte residues on the sample loop (*sample loop memory*) as well as carrier gas contaminations are investigated by (a) while (b) includes analyte residues within the tubing upstream of the sample loop, i.e. stream selection, sample dryer etc. (*system memory*). To get the most complete picture possible, 65 substances were analysed, most of them halo- and hydrocarbons (see supplementary information for a detailed list) on both a HayeSep D as well as a Unibeads 1S sample loop. Substances with low measurement precision (> 10 %) were excluded from the investigation.

In general, most of the detected analyte residues are most probably caused by system contaminations (HFCs from fittings, solenoid valve membranes etc.) or carrier gas contaminations (hydrocarbons) as they show a constant background. In principal, the amount of a residue is dependent on volatility and concentration, so extremely elevated concentrations of low-volatile substances might lead to a memory effect that was not detected in the current investigation with 1 L preconcentration volume of unpolluted ambient air. Detailed results for the two different adsorptive materials tested are discussed in the following.

*Unibeads 1S adsorptive material.* 13 of 65 substances (20 %) did show detectable residues on the sample loop which did not represent a system memory but a system contamination, e.g. from the carrier gas, sealing materials etc. as they were always present and did not disappear in subsequent unloaded injections. Respective residues were generally larger with increasing boiling point (e.g. n-propane < benzene). Most of them were hydrocarbons and the halocarbons chloro- and iodomethane ($CH_3Cl$, $CH_3I$) and chloroethane ($C_2H_5Cl$) as well as HFC-134 ($CHF_2CHF_2$). No further CFCs, HCFCs, PFCs or HFCs were detected in the unloaded sample loop injection (see Obersteiner et al. (2016) for a discussion of detection limits). Of the remaining 52 substances, 36 also did not show any detectable residues in the helium blank. Of the 17 substances that did show residues (contamination and memory effect combined), 7 had residues below 0.5 % of the signal area determined in the preceding ambient air measurement. Again, residues were found mostly for hydrocarbons but not CFCs or HCFCs. Concluding,





the Unibeads 1S sample loop seems to be a good choice for halocarbon monitoring measure-
ments (one measurement per sample) as there were nearly no halocarbon residues in subse-
quent helium blank measurements.
***HayeSep D adsorptive material.*** The HayeSep D sample loop showed a considerably higher
amount of sample loop residues with 22 detectable substances from the selected 65 (34 %).
Again, most of these substances were hydrocarbons but also some halogenated compounds
like Tetrachloromethane ($CCl_4$) and Bromoform ($CHBr_3$). Of the remaining 43 substances, 28
were undetectable in the helium blank (system free of contamination and memory effect). 13
of the detectable substances showed responses of $< 0.5$ % relative to the preceding ambient air
sample, also including CFC-11 with 0.05 % and CFC-113 with 0.2 %. While the named halo-
genated compounds $CCl_4$ and $CHBr_3$ as well as CFC-113 and CFC-11 were undetectable in
subsequent blank gas measurements, residues of many hydrocarbons were persistent, suggest-
ing a system contamination. In summary, the HayeSep D sample loop showed an overall
higher number of residues which is likely caused by a higher desorption temperature of the
Unibeads 1S sample loop which can be heated faster and to a higher temperature without de-
grading the material. Nevertheless, the residues on both adsorptive materials were on a tolera-
ble level (below average measurement precision) for flask measurements with multiple meas-
urements per sample.





# 4 Application

## 4.1 Laboratory operation: flask sample measurements

For quality assurance of the laboratory instrumentation, five air samples were analysed and compared to our reference GC-QPMS (gas chromatograph coupled to a quadrupole mass spectrometer) which uses a similar preconcentration setup (Hoker et al., 2015). Consistent results with the NOAA network (National Oceanic and Atmospheric Administration) were demonstrated for the GC-QPMS in the past during the IHALACE intercomparison (Hall et al., 2014), however with a different sample preconcentration using liquid nitrogen (Brinckmann et al., 2012; Laube and Engel, 2008; Laube et al., 2010). The current laboratory setup using the Stirling cooler-based preconcentration has been described by Hoker et al. (2015) and has shown very consistent results with previous measurements. The samples for the application and intercomparison discussed here were collected between July 7th and September 11th 2015 at Mace Head Atmospheric Research Station in Ireland (53°20′ °N, 9°54′ °W, 30 m above sea level). Samples were filled "moist" (no sample drying) into 2 L electro-polished stainless steel flasks (two flasks in parallel per sampling date). The comparison is extended to include in-situ measurement data from the online monitoring Medusa GC-MS (Miller et al., 2008) operated by the AGAGE (Advanced Global Atmospheric Gases Experiment) network at Mace Head Station. Medusa GC-MS data points were chosen within ±1 hour of the flask samples' sampling time. **Figure 6** shows a comparison of absolute quantification results for CFC-12 ($CCl_2F_2$). Very good agreement within the 1-fold measurement error is achieved in comparison to the Medusa GC-MS and within the 2-fold measurement error in comparison to the reference GC-QPMS. While the Medusa GC-MS is calibrated with secondary calibration gases (AGAGE flasks H-265 and H-266; CFC-12 scale: SIO-05), both our instruments were calibrated with different ternary calibration gasses, referenced to the same secondary calibration gas (AGAGE flask H-218; CFC-12 scale: SIO-05). Taking into account that all three instruments were calibrated with different calibration gases which rely on the same calibration scale but are based on a chain of intercalibrations, this result stands proof for highly accurate measurement results, excluding the absolute scale error.



## 4.2 Aircraft in-situ operation: GhOST-MS

Reliability of operation is best demonstrated with the in-situ GC-MS GhOST-MS[1]. **Figure 7** shows a chromatogram obtained from the injection of a preconcentrated sample volume of 100 mL of ambient air. With a chromatographic runtime of 2.9 minutes and a total cycle time of 4.1 minutes (see also **Table 1**), a data frequency is achieved that is very high for a GC-MS system with a total of 27 identified and simultaneously measured species on m/Q of bromine, chlorine and iodine in negative chemical ionisation mode using argon as reagent gas. The cycle time is limited by cooldown of the adsorptive material (HayeSep D) to −70 °C needed to quantitatively trap the earliest eluting analyte, Halon 1301 ($CBrF_3$). The very good overall performance of the GhOST-MS including the preconcentration unit used in this in-situ application can be inferred from actual measurement data obtained during a research flight of the recent PGS campaign (POLSTRACC/GW-LCycle/SALSA) of the HALO aircraft on flight 160226a (PGS-14). **Figure 8** shows a tracer-tracer correlation between Halon 1301 and Halon 1211 ($CBrClF_2$). The measurements are colour-coded to show potential temperature $\theta$. As expected, the lowest mixing ratios are observed at the highest potential temperature. Both tracers have relatively long steady-state lifetimes of 72 years for Halon 1301 (58-97, derived from model data and observations) and 16 years for Halon 1211 (10-39, model data) (SPARC, 2013) so that a compact correlation of mixing ratios of these two traces gases is expected in the stratosphere (Plumb and Ko, 1992). Due to its relatively low boiling point (−57.8 °C), Halon 1301 is the first species eluting from the chromatographic column. The shape of the chromatographic peak is thus strongly influenced by the injection, as refocusing on the chromatographic column is expected to play a negligible role. As a correlation derived from measurement data can only be as compact as the measurement precision allows, the compactness of the correlation shown in **Figure 8** gives an indication of the high measurement precision achieved with the GhOST-MS. The fact that this compact correlation includes a substance whose precision is strongly influenced by its thermodesorption shows that the sample preconcentration system on GhOST-MS is able to reproducibly trap and desorb even low boiling compounds like Halon 1301.

GhOST-MS has been deployed during a total of more than 200 flight hours on the HALO aircraft without a single failure of the preconcentration unit. In addition, measurements with GhOST-MS were performed as part of the SHIVA campaign in Borneo, providing a complete bromine budget for the upper tropical troposphere up to about 13 km (Sala et al., 2014). The

---

[1] Manuscript on the current GhOST setup and characterisation in preparation by Keber et al.



1    preconcentration unit presented here therefore is not only able to provide high precision but is

2    also able to operate reliably under difficult conditions like aircraft operation with varying hu-

3    midity and temperatures, including operation during humid and hot conditions in the tropics.



## 5 Summary and conclusion

A single-stage, refrigerant-free sample preconcentration unit for ambient air analysis is presented and characterised. The setup has proven to be applicable for both in-situ and laboratory operation and can quantitatively trap and desorb a wide range of halo- and hydrocarbons (see supplementary information). The use of different adsorptive materials is possible with the setup; two of which were used during this work, HayeSep D and Unibeads 1S. Both materials are well suited for analysis of halogenated trace gases in general. While HayeSep D is an established material for this task, Unibeads 1S potentially is a good alternative that has better heat tolerance and showed fewer sample loop blanks in the presented characterisation.

The preconcentration unit is positioned between more sophisticated but also more expensive and complicated solutions like e.g. the Medusa preconcentration unit described by Miller et al. (2008) and setups that use less powerful, Peltier-based cooling options that sacrifice adsorption temperature and therefore reduce the trappable substance range. The described setup is unique in terms of the used cooling technique, a Stirling cooler. The latter allows very low temperatures of −120 °C tested in this work and −173 °C reported by Eyer et al. (2016) for the preconcentration of methane with a comparable Stirling cooler without having to rely on a cooling agent like liquid nitrogen or liquid argon. The Stirling cooler as a cooling option is ideally suited for in-situ, remote-site operation, where refrigerant-based cooling options are very difficult to operate and space is limited – like the aircraft-based in-situ GC-MS instrument GhOST-MS. Moreover, the absence of mechanical/moving parts as well as the lack of necessity of vacuum insulation of cooled parts facilitates installation and maintenance. No exchange of adsorption tubes is necessary. Overall, the setup is relatively cheap with the Stirling cooler being the most expensive part by far.

The simplicity of the single-stage design also has a downside; a major problem is the trapping of large amounts of $CO_2$ and injection into the detection system (see also section 3.2), especially when using trapping temperatures below -80 °C. Due to this limitation, the current configuration is not applicable to highly volatile compounds like $CF_4$, $C_2F_6$ or $C_2H_6$. Cooling capacity should however be sufficient to ensure quantitative trapping of such compounds on a suitable adsorptive material. Therefore, a starting point for future improvement is removal of $CO_2$ to extend the already large substance range by compounds of higher volatility. Regarding desorption, no blank residues were found for halocarbons that would cause concern or render the setup unsuited for halocarbon analysis (see "Appendix B: Blank Residues"). However,



relatively large amounts of hydrocarbons remained in blank measurements. These blanks are
not an inherent problem of the preconcentration setup but more likely due to the adsorptive
materials used. Additional experiments are needed to reduce those uncertainties and extend
the applicability of the preconcentration unit to quantitative hydrocarbon analysis.





## 1 Acknowledgements

This work was supported by research grants of the Deutsche Forschungsgemeinschaft (DFG),
EN367/12-1 (*FASTOF*), EN367/5-2 (*GhOST-MS*) and EN367/13-1 (*PGS*). We thank L. Mer-
kel and the workshop of the institute for their contribution of technical drawings and compo-
nent construction. Special thanks go to G. Spain for sample collection at Mace Head Station
as well as the PGS campaign team lead by H. Oelhaf and B.-M. Sinnhuber which gave us the
opportunity to create a set of excellent in-situ measurement data with the GhOST-MS.



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





**Tables**
**Table 1.** Cycle times at $T_A$ of -80 °C / -120 °C (laboratory setup) and -70 °C (in-situ setup), based on
operational data. Laboratory setup configuration: Sunpower CryoTel CT Stirling cooler, preconcentra-
tion volume: 500 mL at 100 mL·min$^{-1}$, $T_D \approx$ 200 °C for 3 min. In-situ setup configuration: Twinbird
SC-TD08 Stirling cooler, preconcentration volume: 100 mL at 100 mL·min$^{-1}$, $T_D \approx$ 200 °C for 1 min.
Adsorptive material, both setups: HayeSep D. Due to a smaller coldhead, cooling rate and warm-up
during desorption are considerably larger with the in-situ setup, despite the shorter desorption time.

| $T_A$ [°C] | cooling rate at $T_A$ [°C·min$^{-1}$] | warm-up during desorption [°C] | minimum cycle time including preconcentration after $T_A$ is reached [min] |
|---|---|---|---|
| Laboratory instrument (GC-TOFMS) | | | |
| −80 | −2.2 | 7.7 | 8.5 |
| −120 | −1.2 | 16.3 | 18.6 |
| In-situ instrument (GhOST-MS) | | | |
| −70 | −4.1 | 13.5 | 4.1 |



**Table 2**. Results from a volume variation experiment, comprising measurements of the same reference
air with preconcentration volumes (PrcVol) of up to 2, 5 and 10 L. Laboratory setup, adsorptive mate-
rial Unibeads 1S. Volume-corrected detector response is referenced against calibration preconcentra-
tion volumes of 1 L (rR). rR <100% indicates underestimation (e.g. loss by breakthrough); rR >100%
indicates overestimation (e.g. increase by a memory effect from the preceding sample or contamina-
tion). Breakthrough is observed for COS at a preconcentration volume of 10 L while ethyne shows
signs of a system contamination (rR >100% despite a higher volatility compared to COS). CFC-12 and
CFC-11 show no indication of breakthrough, with all deviations from 100% rR below 3 σ.

| Substance | PrcVol [L] | rR | rR: 1 σ | PrcVol [L] | rR | rR: 1 σ | PrcVol [L] | rR | rR: 1 σ |
|---|---|---|---|---|---|---|---|---|---|
| Ethyne ($C_2H_2$) | | 102.0% | 0.66% | | 108.9% | 0.70% | | 109.2% | 0.70% |
| Carbonyl sulfide (COS) | 2 | 102.2% | 0.82% | 5 | 100.9% | 0.81% | 10 | 64.8% | 0.52% |
| CFC-12 ($CCl_2F_2$) | | 99.9% | 0.41% | | 100.7% | 0.42% | | 100.6% | 0.42% |
| CFC-11 ($CCl_3F$) | | 100.2% | 0.21% | | 100.5% | 0.22% | | 100.6% | 0.22% |



**Table 3.** Retention times $t_R$ with relative standard deviations rSD and variances in [s] for selected sub-
stances (same as **Table 2**) as well as their respective average signal width expressed as FWHM in [s].
Values derived from 112 individual measurements of different ambient air samples using the ramped
GC program. Sample loop adsorptive material: HayeSep D. HFC-23 is the first detectable substance,
least separated by chromatography. CFC-11 can be considered a reference for optimal chromatograph-
ic performance of the given setup.

| Substance | $t_R$ [min] | $t_R$ rSD | Variance [s] | Avg. Peak Width [s] |
|---|---|---|---|---|
| HFC-23 ($CHF_3$) | 3.01 | 0.105% | 0.0386 | 4.09 |
| Ethyne ($C_2H_2$) | 3.74 | 0.047% | 0.0118 | 2.77 |
| Carbonyl sulfide (COS) | 3.86 | 0.040% | 0.0092 | 2.29 |
| CFC-12 ($CCl_2F_2$) | 5.01 | 0.014% | 0.0018 | 2.26 |
| CFC-11 ($CCl_3F$) | 7.25 | 0.006% | 0.0008 | 2.24 |



**Figures**

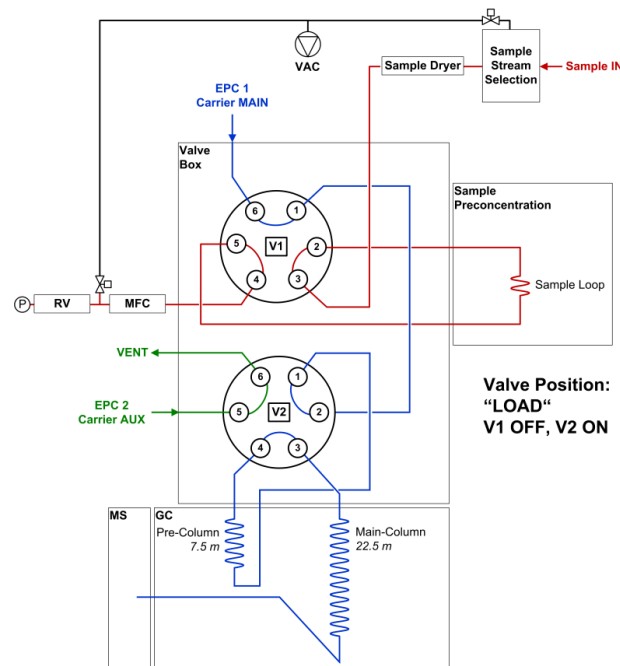

**Figure 1.** Flow scheme showing the gas flow during preconcentration. Two electronic pressure
controllers, EPC 1 and EPC 2, control the carrier gas flow. The two 6-port 2-position rotary
valves V1 and V2 are set to OFF/ON position. A sample is preconcentrated (red flow path);
sample components not trapped in the sample loop flow through the mass flow controller
(MFC) into the reference volume (RV). By switching V1 to ON position (for desorption), the
sample loop is injected onto the GC column. Sample loop as well as reference volume and
stream selection valves are evacuated prior to the preconcentration of the next sample. By
switching V2 to OFF, it separates pre- and main-column; the pre-column is flushed backwards.
This prevents high-boiling, non-targeted species from reaching the main-column.





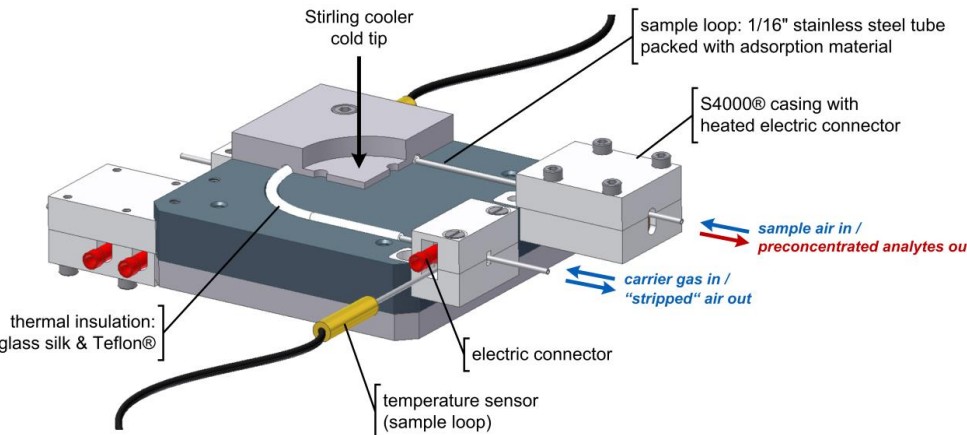

**Figure 2.** Technical drawing of the coldhead and sample loop placed inside. Three plates of
anodized aluminium can hold two sample loops. The Stirling cooler's cold tip screwed to the
coldhead removes heat for cooling. Heat for sample desorption is generated by a current directly
applied to the sample loop. The electric connector in the direction of sample flow (upper right
side of the drawing) is heated constantly to 150 °C to avoid a cold point due to the mass of the
electric connector and its proximity to the coldhead (S4000® insulation material:
Brandenburger, Germany).





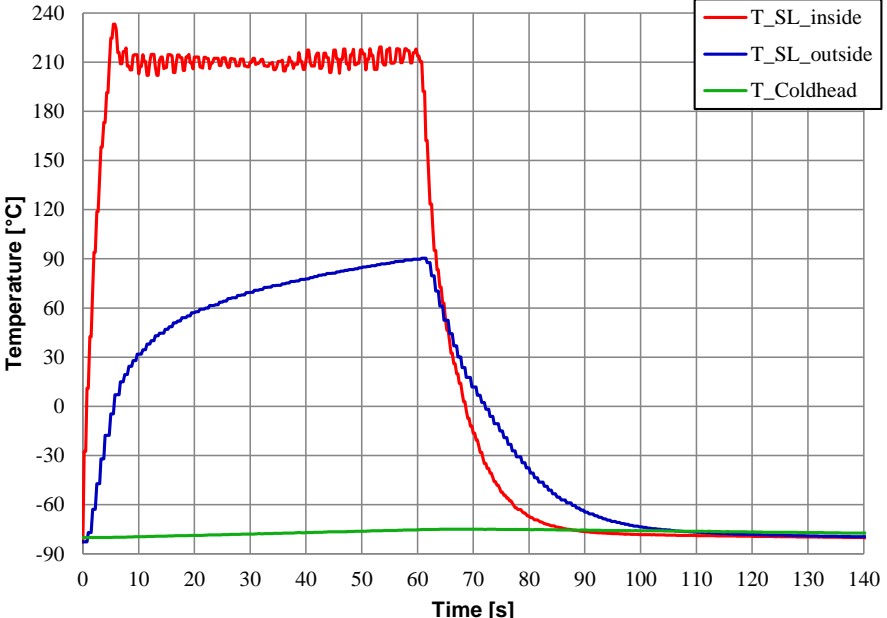

**Figure 3.** Desorption temperature curve inside the sample loop with a preceding adsorption
temperature of −80 °C and a subsequent cool-down from desorption to adsorption temperature.
Red curve, "T_SL_inside": signal from temperature sensor shifted inside the sample loop. Blue
curve, "T_SL_outside": temperature sensor signal from the sensor welded to the outer sample
loop tubing wall. Green curve, "T_Coldhead": temperature of the coldhead. Deterministic
heater, output in this example: 50 % in stage 1, held 5 s, and 30 % in stage 2, held 55 s. The
periodic oscillation of $T_D$ observed is a result of a very slow pulse width modulation used in the
testing setup: 100 ms period with 10 ms minimum increment.



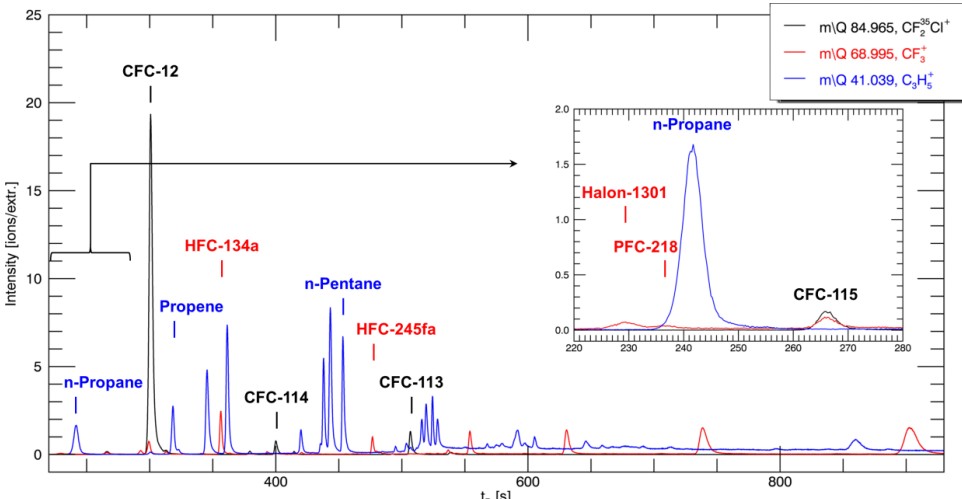

**Figure 4.** Chromatogram from a 1 L ambient air sample obtained with the GC-MS setup
described in Obersteiner et al., 2016. X-axis: retention time $t_R$ in seconds. Y-axis: signal
intensity expressed as ions per extraction which are derived from a 22.7 kHz TOFMS extraction
rate, averaged to yield a mass spectra rate of 4 Hz. X- and Y-axis description also valid for the
magnified section. Black graph: mass-to-charge ratio (m/Q) = 84.965 signal from a typical CFC
fragment ion $CF_2{}^{35}Cl^+$. Red graph: m/Q = 68.995 signal from a typical PFC or HFC fragment
ion $CF_3{}^+$. Blue graph: m/Q = 41.039 signal from a typical hydrocarbon fragment ion $C_3H_5{}^+$. The
magnified section shows the chromatographic peak of n-propane and three other compounds to
demonstrate injection quality of substances least re-focused by chromatography.





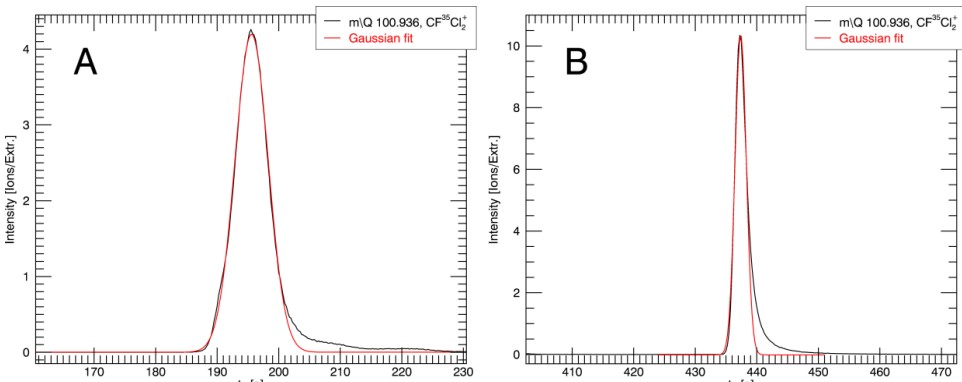

**Figure 5.** Comparison of chromatographic peak shapes of the $CF^{35}Cl_2^+$ fragment ion signal of
CFC-11 ($CFCl_3$), from an injection of 1 L preconcentrated ambient air onto the GC column kept
isothermal at 150 °C (A) and onto the GC column kept at 45 °C and ramped to 200 °C
subsequently (B) (see section 3.1). X-axis: retention time $t_R$ in seconds; $t_R$ interval shown is 70 s
in both plots. Y-axis: signal intensity expressed as ions per extraction (see **Figure 4**). The red
curve shows a Gaussian fit for comparison of actual peak shape and a peak shape that is
considered ideal. FWHM of fit: (A) 6.3 s (0.10 min) and (B) 2.0 s (0.03 min). Adsorptive
material: Unibeads 1S.





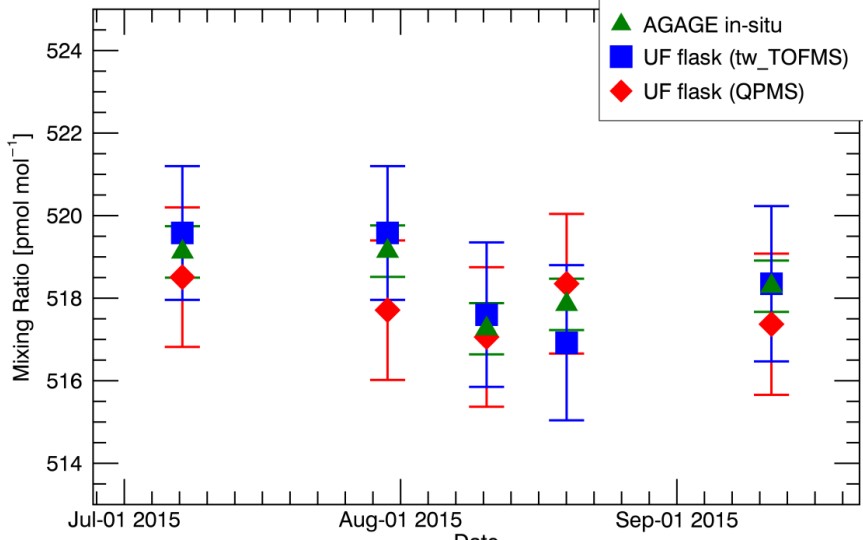

**Figure 6.** CFC-12 (CCl$_2$F$_2$) mixing ratios at Mace Head Atmospheric Research Station, Ireland
(53°20′ °N, 9°54′ °W, 30 m above sea level) derived from 2 L stainless steel flask samples
measured with the instrument in description (GC-TOFMS, blue squares), our reference
instrument (GC-QPMS, red diamonds) and values taken from the online measurement data of
the in-situ Medusa GC-MS (green triangles). Error bars: 1-fold measurement precision of each
instrument (Medusa system: typical precision taken from Miller et al. (2008)). Calibration scale,
all instruments: SIO-05.

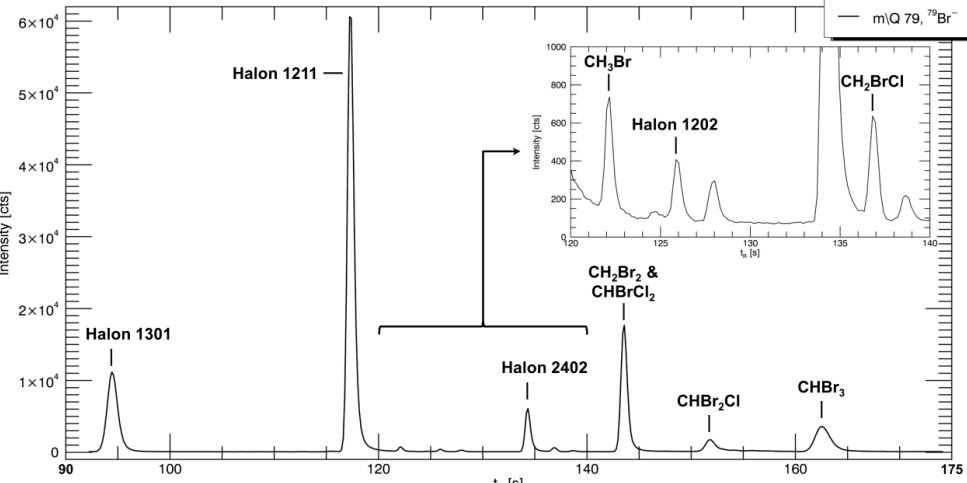

**Figure 7.** Chromatogram from a preconcentration of 0.1 L ambient air obtained with the in-situ
GC-MS setup GhOST-MS. X-axis: retention time $t_R$ in seconds. Y-axis: signal intensity in
counts, arbitrary unit. MS: Agilent 5975C in negative chemical ionization mode (reagent:
argon). Black graph: mass-to-charge ratio $m/Q = 79$ signal of $^{79}Br^-$ ions from brominated trace
gases.





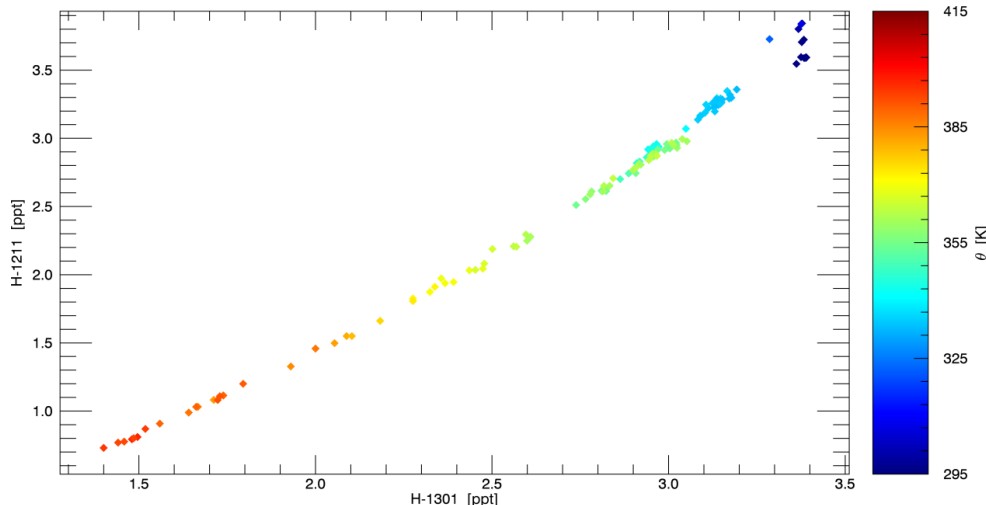

**Figure 8.** Tracer-tracer correlation of Halon 1301 (CBrF$_3$, x-axis) vs. Halon 1211 (CBrClF$_2$,
y-axis). Color code indicates potential temperature $\theta$ in [K]. Data was obtained during the
POLSTRACC mission with the HALO aircraft, flight 160226a (PGS-14). Preliminary data;
calibration scale of Halon 1301 and 1211: SIO-05.