# Peer review of "A versatile, refrigerant-free cryofocusing-thermodesorption"

_Atmospheric Measurement Techniques, 2016_

## Referee Comment (RC1) · Anonymous Referee #1 · 7 Jul 2016

General comments:

The manuscript of Florian Obersteiner and co-authors with the title "A versatile, refrigerant-free cryofocusing-thermodesorption unit for preconcentration of traces gases in air" presents a temperature swing adsorption unit for halogenated trace gases. A basic description of the setup of the developed preconcentration unit was already given in Obersteiner et al. AMT (2016) doi:10.5194/amt-9-179-2016 and other manuscripts of the authors. Nonetheless the present manuscript adds details on the instrument design, optimisation, performance characteristics and application examples. The manuscripts give strengths and weaknesses of the novel design as uneven temperature distribution of the trap or problems in getting reliable temperature data.

Research on trace gases, e.g. halogenated components or GHG isotopologues is very active and the manuscript is therefore timely and of high interest for readers of AMT and potential future users of this technique. I have some concerns on the consistency of the structure of the manuscript, which could be improved as detailed below. In addition, the wording is sometimes colloquial and might be improved. I have a number of suggestions for technical corrections the authors might consider to improve the consistency and readability of the manuscript.

In summary, I suggest publication in Atmospheric Measurement Techniques with minor revisions.

Specific comments:

SI units should be used throughout the manuscript, e.g. "K" instead of "°C" for temperature.

Page 1 Line 25: The term "injection quality" is arbitrary and might be replaced.

Page 4 – 8 Section 2 Instrumentation: The main focus of this section is on the technical description of the preconcentration device including hardware setup and preconcentration procedure. However, details on the hardware are partly missing, partly they are given in section 3 Characterisation. Please give all relevant technical information on components implemented in the preconcentration device including information on model, manufacturers etc. in section 2, where it is mentioned for the first time.

Page 4 Line 7-8: The sentence "A preconcentration system can only be as good as the analytical set-up behind it" is colloquial and might be deleted or rephrased. The overall performance of an analytical system might be either limited by the preconcentration unit or the analyser.

Page 4 Line 13 ff: The title of the section is "loose" and might be change to something like "Setup of the preconcentration device and integration for GC applications". In this section the preconcentration device is described in detail for the first time and therefore reference to Figure 1 and 2 should be given. Technical information on all components in the preconcentration device with model number, manufacturer etc. should be added.

Page 4 Line 14: Please give details on the applied adsorbents.

Page 4 Line 19: The term "warm column" is imprecise and might be replaced or just deleted as details are given thereafter.

Page 4 Line 20 - 21: Please rephrase the sentence to something like: The system does not involve a separate refocussing procedure as used in other preconcentration systems (Miller et al. 2008) but the analytes are directly purged onto the GC column for further separation.

Page 4 Line 24 - 27: Please give details on the applied MFC. It is unclear how the MFC could be used for large sample volume, could it be: The MFC provides an alternative way to quantify the sample volume. Thereby, operation of the MFC without the downstream reference tank adds flexibility with respect to sample volumes larger than the reference tank.

Page 5 Line 6: The term "in this case" might be rephrased. I assume the two sample loops are not equivalent regarding temperature as one is closer to the Stirling cooler than the other one, which complicates the comparison of temperature data from both loops. How can these temperatures be compared – please comment on this?

Page 5 Line 19 – 22: Please give information on the Stirling cooler used in the presented work.

Page 5 Line 25 – 29: This statement that an "idle time" is needed to reach $T_A$ after desorption is abstract as Figure 3 shows a constant cold plate temperature. From other statements later in the text I assume that at lower cold plate temperatures (e.g. -120°C) heating the trap during desorption affects the cold plate temperature – please comment on this?

Page 5 Line 27 – 28: The sub-sentence " … is transferred to the coldhead as the sample loop is kept directly inside with only …" might be rephrased.

Page 6 Line 10: Please give a number for the "slightly higher $T_A$".

Page 6 Line 4 – 7 and Line 15 – 18: There seems to be a discrepancy between the statements on cooldown time given in Line 4 – 7 (18.6 vs. 8.5 minutes) and in Line 15 – 18 (90 vs. 30 s) for different $T_A$ of -120°C versus -80°C – Please comment?

Page 7 Line 16 ff: Measurement of the trap temperature and thus its control seems to be difficult with a sensor on the trap surface as it is affected by the cold plate. Would it be possible to use the sensor mounted within the second trap to control the temperature of the first although the traps are not identical?

Page 9 section 3: Would a title similar to "performance characteristics" be better?

Page 9 section 3.1: As detailed above to improve readability detailed information on the preconcentration unit (e.g. Stirling cooler, MFC) should be given in section 2 and deleted here. Section 3.1 should mainly contain information on the specific set-up used to determine "performance characteristics" as breakthrough volume etc..

Page 10 Line 19: The wording "stripe the air of nitrogen etc." should be rephrased.

Page 11 Line 8: In Table 2 the relative response is given in "%" please use consistent terms.

Page 13 – Page 14: Different wording are used to differentiate detrimental contamination effects: sample loop memory, system memory and system contamination. Please unify and simplify the wording and the ways how to test them, if possible beginning of the section 3.3.

Page 14 Line 18: "Concentration in the previous run"

Page 16 Line 3 ff: It would be good to state first what the inter-comparison is based on: The laboratory based instrumentation (GC-TOFMS) is compared to GC-QPMS and GC-MS. As GC-TOFMS and GC-QPMS are based on a similar preconcentration setup, and the preconcentration setup should be tested, it is unclear what we can learn from that.

Page 17 Line 14: Please give details on the argumentation related to the "potential temperature".

---

## Referee Comment (RC2) · B. R. Miller (Referee) · 17 Jul 2016

Some general comments: This manuscript describes the relatively simple design of a custom-designed preconcentration unit, which employs a Stirling cooler as the cold source for cryogenic preconcentration of trace atmospheric compounds. The basic design is common to several of their instruments, results of which have been published recently. The lack of need for an insulating vacuum chamber for these cold parts is an appealing characteristic as it greatly simplifies design, reduces cost and weight. The rapid analysis time that they have achieved also makes this a desirable field and laboratory instrument. The focus of the current manuscript is a detailed description and characterization common to these instruments, going beyond what was published

in Hoker et al., 2015, Obersteiner et al., 2016 and Sala et al., 2014. The authors do present in-depth results of the chromatographic peak shape resulting from cryofocus and subsequent desorption of analytes. They also discuss results from analyte break-through tests and report some of their findings of system contamination.

I appreciated the depth of investigation of the 'analyte residues' issue. In such preconcentration methodologies, and at such low analyte mole fractions, these contamination issues can be a very difficult problem to investigate and rectify. So by sharing their observations, the authors are contributing to a knowledge base that may someday be useful in developing some remedies.

To add to the value of this residue investigation, I would like to see them include, for example in Table 4, which analyte contaminations they attribute to 'carry-over' from a previous sample, versus those derived from system hardware (e.g., outgassing of polymers, or desorption from surfaces), versus those derived from the carrier gas. In the text, they do allude to such finding with regard to a select set of analytes, but stop short of specification for each of the dozens of compounds listed in the table. Undoubtedly, there will be some analytes for which attribution is not readily discernible, but I would hope that they are able to elaborate with more useful details regarding more clear-cut cases.

Results from Table 4 present some difficulty in interpretation. For example, iodomethane shows a HayeSep D residue of 43.9%. This is, as defined in the caption, the ratio of the blank gas response to the preceding 1 L air sample, expressed as a percentage. The blank gas injection is defined as an injection of purified helium. From this information, we cannot deduce whether this high percentage is attributable to carry-over of analyte from the air injection into the subsequent blank injection, or if the carrier gas is contaminated with an iodomethane-like substance or if some hardware upstream from the adsorbent tube is outgassing or desorbing the contamination.

Further difficulty in interpreting table 4 stems from the precision limitation that they im-
pose. The authors exclude analytes from Table 4 that show "...show poor precision  $\geq$  10%." With a 43.9% residue of unknown origin, how is a

attempt analysis of analytes which elute under the CO2. On their 20 mg of adsorbent, chilled to -80 C or -120 C, they need to show that the co-adsorption of CO2 does not adversely affect adsorption and/or desorption of their analytes. Consider that a 1-L sample of ambient air has order 0.4 ml STP of pure CO2, enough gas to fill a 50 cm length of their 1 mm I.D. tubing. While it is unlikely that they have quantitative adsorption of all the CO2, we aren't given any evidence that sufficient CO2 does not remain to disrupt the desorption of the other analytes and/or disrupt the sensitivity of the detector. In instruments in which no CO2 mitigation is performed, I have observed the co-desorption of CO2 to cause peak width and shape changes in certain co-adsorbed analytes. I have also observed short-term MS detector response reduction due to injection of CO2-containing samples, with subsequent 'rebound' in sensitivity with CO2-free samples. An experiment that the authors could perform to examine this detector issue is to sequence injections of CO2-containing and CO2-free air-like mixtures mixed in with injections of a reference standard gas. If the amount of injected CO2 is having an effect on the reference gas sensitivity, it will be apparent when the sample preceding the reference contained or did not contain CO2.

Table 2 and corresponding text present unbelievably precise chromatographic retention times, given that this represents 112 injections of various samples. Are these the correct units? Taking CFC-11 for example, 0.006%/100\*7.25s\*60s/min yields 0.19s standard deviation for the CFC-11 retention time. Also, if the fourth column truly is a variance, then the units would be the square of the measurement units, i.e., sec(sup)2. Same applies to line 25 of page 13, where the units would be sec(sup)2.

Retention times vary over time scales on the order of a week in such instruments due to several factors. First, their retention times should vary inversely with ambient pressure. Their electronic pressure control (EPC) of their gas chromatograph references atmospheric pressure such that as ambient pressure increases, the chromatographic pressure gradient between EPC and the absolute vacuum of the detector increases, causing retention time to decrease. Another run-to-run variable is the flow resistance
of the gas in the heated adsorbent tube. Figure 3 illustrates quite large temperature oscillations during desorption. Are these oscillations really so reproducible that the gas viscosity is identical in each run?

The authors are commended for making the substance breakthrough experiments. However, their conclusions regarding "quantitative trapping" (e.g., page 11, line 8) are only valid if one assumes that the instrument is linear in response, an assumption for which no evidence is provided. For example, the greater than 100% response ratio of ethyne may be attributable to "memory effect" as they claim, but the 0.3% to 1.4% 'residue' for ethyne listed in table 4 does not support the observed 109% 'breakthrough result' reported in table 2. There are more definitive tests for characterizing break-through, several of which are described in the 1999 thesis work of Dr. Brian Greally from the University of Bristol, UK. Use of the technique of varying the volume of sample gas is one method to examine system non-linearities, but this must be accompanied by a confirming technique such as analysis of a suite of air-like gases of differing analyte mole fraction.

**Specific comments:**

The authors refer to their micro-packed adsorbent tube as a 'sample loop', which I find an unusual use of the term. The use of the term 'sample loop' in the general literature usually refers to an open tubular device which is used to measure the sample volume, typically by pressure measurement under fixed temperature and volume assumptions. I would suggest using a more common terminology for their adsorbent tube.

Pg. 3, line 9: LN2 and Ar used for cooling in these kinds of instruments are not typical examples of 'refrigerants', but of cryogens. Refrigerants undergo expansion (cooling) and subsequent recompression cycles.

Pg. 4, line 28: "self-written control software" should be changed to "custom-written" or similar.
Pg. 10, line 19: "...strips the air of its most abundant constituents..." makes it sound like you are preconcentrating N2, O2, etc., and not the trace target compounds.

Pg. 14, line 16: "In general, most of the detected analyte residues are most probably caused by system contaminations (HFCs from fittings, solenoid valve membrane etc.) ..." By 'solenoid valve membranes, are you referring to those in the EPCs?

Table 4 caption: Why is it that some analytes that show large residues, e.g., iodomethane at 43.9% for HayeSep D, are included in this table when the caption states that those that "...show poor precision  $\geq$  10% were excluded?"

Figure 4 caption and elsewhere: "n-propane" should be simply "propane".

---

## Referee Comment (RC3) · Anonymous Referee #3 · 20 Jul 2016

In this article Obersteiner and co-workers present a novel device to sample highly volatile compounds without the need of a cooling agent. Although it has been designed for two specific gc-ms systems, its use in many applications, where cooling agents are not available, is conceivable. The device has been very thoroughly characterized very and its strengths and weaknesses are described in the paper. I only have some remarks to improve the structure and the readability of the publication.

I would suggest publishing the article with minor revisions in Atmospheric Measurement Techniques.

Page 2, Line 13: The term 'direct measurement' is not unambiguous., I would rephrase this sentence.

[Figure]

Page 3, Line 3 Is there literature on the PerkinElmer system?

Page 4, Line 19 Write flushed in direction opposite to sampling flow (instead of backwards)

Page 5, Line 30 The term chromatographic runtime is not clear to me. Does it mean that the chromatographic separation is the time limiting step?

Page 6, Line 10 Use time resolution (instead of number of measurements per time)

Page 6, Line 16 reach (instead of reached)

Page 6, Line 11 Why should an increased preconcentration flow increase the time resolution? Is it due to the decreased size of the cold head?

Page 6, Line 18 The difference of total run time, overall cycle times and minimum cycle time is not clear to me. Is the time resolution of e.g. the GhOST-MS equivalent the minimum cycle time plus the time for cool-down, or is the cool-down phase already during the GC runtime? You write that the sample-loop cool down time is not a limiting factor. But on page 17 line 8 you write that the cycle time of the GhOST-MS is indeed limited by the cooldown of the adsorptive material. Isn't it the cooldown time? Maybe a schematic diagram of the runtime with the cool down-time, the desorption time and the chromatographic runtime could help to avoid any confusion on the different cycle times.

Page 7, Line 27 A short description of the 'stages' should be already included here.

Page 10, Lines 7-14 The importance of desorption for the chromatographic peak shapes is very nicely discussed in chapter 3.3. Hence, Figure 4 should be discussed in chapter 3.3 and not already here.

Page 10 Line 20 When atmospheric ozone is trapped in the cryofocus it can degrade the alkenes mentioned in the supplement. Has the influence of ozone on the recovery of VOCs been investigated?

[Figure]

Page 11, Line 1 How is the back pressure of 2.5bar generated? From figure 1 the sample loop should have ambient or reduced pressure during sampling.

Page 13, Line 14 Peak tailing is by definition on the right flank, so I would propose to simply write peak tailing or tailing.

Page 13, Line 30 You write that the tailing effect could potentially be reduced by re-focusing the high-volatile analyte fraction on a second sample loop. How can this be achieved? Do you need a third valve for it, or would you place it in front of the main column?

---

## Author Comment (AC1) · 8 Sep 2016

**Answers to the Reviewers, amt-2016-196**

Dear Referees, we thank you very much for the time and effort you put in reading and commenting our manuscript. We hope that we can address all your comments in a satisfying manner by additional explanations and changes to the manuscript. All your comments are listed in this document, followed by our response. Comments are in italics, our answers in standard font. Red font colour marks text added to or changed in the manuscript. Please find our…

- answers to referee #1 on p. 2 ff.,
- answers to referee #2 on p. 13 ff.,
- answers to referee #3 on p. 28 ff.,
- and further changes to the manuscript on p. 35 ff.

The mark-up version of the revised manuscript can be found on p. 36 ff. of this document.

on behalf of all authors,

Florian Obersteiner

**Comments of referee #1**

*General comments:*

*The manuscript of Florian Obersteiner and co-authors with the title "A versatile, refrigerant free cryofocusing-thermodesorption unit for preconcentration of traces gases in air" presents a temperature swing adsorption unit for halogenated trace gases. A basic description of the setup of the developed preconcentration unit was already given in Obersteiner et al. AMT (2016) doi:10.5194/amt-9-179-2016 and other manuscripts of the authors. Nonetheless the presented manuscript adds details on the instrument design, optimisation, performance characteristics and application examples. The manuscripts give strengths and weaknesses of the novel design as uneven temperature distribution of the trap or problems in getting reliable temperature data.*

*Research on trace gases, e.g. halogenated components or GHG isotopologues is very active and the manuscript is therefore timely and of high interest for readers of AMT and potential future users of this technique. I have some concerns on the consistency of the structure of the manuscript, which could be improved as detailed below. In addition, the wording is sometimes colloquial and might be improved. I have a number of suggestions for technical corrections the authors might consider improving the consistency and readability of the manuscript.*

*In summary, I suggest publication in Atmospheric Measurement Techniques with minor revisions.*

*Specific comments:*
* * *
***1 -*** *SI units should be used throughout the manuscript, e.g. "K" instead of "°C" for temperature.*

➢ The reviewer is correct in pointing out that temperature is not shown in SI unit but degrees Celsius. We decided to do so as °C is more common in gas chromatography and also sample preparation. For consistency, we use only this unit throughout the manuscript, although K could of course be used to express e.g. temperature differences. Because K would be cumbersome in many places, °C is, due to the easy conversion, equally appropriate for the understanding of the manuscript and we therefore suggest not to change the unit.
* * *
***2 -*** *Page 1 Line 25: The term "injection quality" is arbitrary and might be replaced.*

➢ We rephrased the sentence accordingly: "We present the instrumental design of the preconcentration unit and demonstrate capabilities and performance through the examination of analyte breakthrough during adsorption, repeatability of desorption and analyte residues in blank tests."

*3 - Page 4 – 8 Section 2 Instrumentation: The main focus of this section is on the technical description of the preconcentration device including hardware setup and preconcentration procedure. However, details on the hardware are partly missing; partly they are given in section 3 Characterisation. Please give all relevant technical information on components implemented in the preconcentration device including information on model; manufacturers etc. in section 2, where it is mentioned for the first time.*

> ➤ An earlier version of the manuscript was actually structured like the reviewer suggests – however, we then changed the structure to the current version as the idea was to give a principal description first and then give an example (including detailed technical information) which should stand exemplary for the preconcentration setup we describe.

> ➤ To make that point clearer, we
>> ○ … changed the section title to Implementation of cryofocusing and thermodesorption
>> ○ … restructured and extended the introduction of sect. 2: "[…] our analytical setups presented in Sala et al., (2014), Hoker et al., (2015) and Obersteiner et al., (2016). Technical details are listed in Table 1 for all three setups we operate. The following section 2.1 outlines the general measurement procedure and gas flow as well as its integration into a chromatographic detection system. Sections 2.2 and 2.3 describe the implementation of the main operations of the unit; cooling ("trapping", i.e. preconcentration of analytes) and heating (desorption of analytes). A preconcentration system can always only be as good as the analytical set-up behind it. The preconcentration system described here was designed for the coupling with a chromatographic system but in principle could also be adapted for coupling with other techniques. Specific technical components of the instrumentation used in this work to characterise the preconcentration unit will be listed in section 3."
>> ○ … added an overview table (Table 1) with relevant technical details on all three preconcentration setups we built; Sala et al. 2014, Hoker et al. 2015 and Obersteiner et al. 2016.

***4 -*** *Page 4 Line 7-8: The sentence "A preconcentration system can only be as good as the analytical set-up behind it" is colloquial and might be deleted or rephrased. The overall performance of an analytical system might be either limited by the preconcentration unit or the analyser.*

➤ Deleted the sentence as the reviewer suggested.

***5 -*** *Page 4 Line 13 ff: The title of the section is "loose" and might be change to something like "Setup of the preconcentration device and integration for GC applications". In this section the preconcentration device is described in detail for the first time and therefore reference to Figure 1 and 2 should be given. Technical information on all components in the preconcentration device with model number, manufacturer etc. should be added.*

➤ Rephrased the title as suggested: Preconcentration procedure and integration for GC application. A reference to Fig. 1 is already given in sect. 2.1; a reference to the technical drawing of the coldhead seems a bit misleading to us, as this part of sect. 2 is not dedicated to this specific part of the setup.
➤ Regarding technical details, please refer to our answer on comment #3.

***6 -*** *Page 4 Line 14: Please give details on the applied adsorbents.*

➤ Please refer to our reply on comment #3.

***7 -*** *Page 4 Line 19: The term "warm column" is imprecise and might be replaced or just deleted as details are given thereafter.*

➤ Deleted as suggested.

***8 -*** *Page 4 Line 20 - 21: Please rephrase the sentence to something like: The system does not involve a separate refocussing procedure as used in other preconcentration systems (Miller et al. 2008) but the analytes are directly purged onto the GC column for further separation.*

 ➢ Rephrased the sentence accordingly: The system does not involve a refocussing procedure as implemented in other preconcentration systems (Miller et al., 2008). The analytes are purged directly onto the GC column for separation.

***9 -*** *Page 4 Line 24 - 27: Please give details on the applied MFC. It is unclear how the MFC could be used for large sample volume, could it be: The MFC provides an alternative way to quantify the sample volume. Thereby, operation of the MFC without the downstream reference tank adds flexibility with respect to sample volumes larger than the reference tank.*

 ➢ The reviewer is exactly right with his assumption. The MFC integrates the sample flow and gives accurate results for volumes greater approximately 100 mL as we derived from a comparison experiment with the pressure sensor. The integration of smaller volumes seems to be inaccurate (underestimation), which is why we kept the reference volume tank installed.
 ➢ Rephrased the sentence: The MFC can also be used to determine the sample volume and thereby adds flexibility with respect to sample volumes larger than the reference tank.

**10 -** *Page 5 Line 6: The term "in this case" might be rephrased. I assume the two sample loops are not equivalent regarding temperature as one is closer to the Stirling cooler than the other one, which complicates the comparison of temperature data from both loops. How can these temperatures be compared – please comment on this?*

➢ Provided that the ambient temperature insulation of the coldhead is intact, we actually found no significant difference between both temperature sensor values (type Pt100; $\Delta T \sim 1$ K), likely due to the high thermal conductivity of the aluminium coldhead. We observed an increase of this temperature difference if less coldhead insulation is applied.

➢ Added a sentence to the manuscript to acknowledge this fact, p.5,l.8: "Provided that the coldhead insulation is sufficient and intact, no significant temperature differences occur between both traps due to the high heat conductivity of the aluminium".

**11 -** *Page 5 Line 19 – 22: Please give information on the Stirling cooler used in the presented work.*

➢ Please refer to our answer on comment #3.

**12 -** *Page 5 Line 25 – 29: This statement that an "idle time" is needed to reach TA after desorption is abstract as Figure 3 shows a constant cold plate temperature. From other statements later in the text I assume that at lower cold plate temperatures (e.g. -120°C) heating the trap during desorption affects the cold plate temperature – please comment on this?*

➢ The reviewer is right in pointing out that the y-scale of Fig. 3 makes it difficult to observe the rising temperature of the coldhead. There actually is a rise of ~4.5 K.
  o We therefore added a second y-axis to display coldhead temperature and illustrate the point.

➢ As the thermal insulation between sample loop and coldhead is constant, heat flow increases if the temperature difference between coldhead and sample loop increases (thermodesorption at equal $T_D$ but lower $T_A$). More heat flows into the coldhead and it therefore gets warmer.

**13 -** *Page 5 Line 27 – 28: The sub-sentence " … is transferred to the coldhead as the sample loop is kept directly inside with only …" might be rephrased.*

➢ Rephrased the sentence to make the point clearer: […] a certain amount of heat flows from the sample loop into the coldhead as the sample loop is kept directly inside with only a small amount of insulation.

➢ Please note: the term "sample loop" was changed to "trap" or "preconcentration trap" in response to comment #12 of referee #2.
* * *
**14 -** *Page 6 Line 10: Please give a number for the "slightly higher TA".*

➢ Added the info to the sentence, (~ −72 °C)
* * *
**15 -** *Page 6 Line 4 – 7 and Line 15 – 18: There seems to be a discrepancy between the statements on cooldown time given in Line 4 – 7 (18.6 vs. 8.5 minutes) and in Line 15 – 18 (90 vs. 30 s) for different TAof -120°C versus -80°C – Please comment?*

➢ The reviewer is correct in pointing out that these statements are given imprecisely. What is meant in the first statement is the total time of a measurement cycle. The second statement refers to the time needed until sample loop and coldhead reach equal temperatures after thermodesorption. This does not imply that $T_A$ is also reached after that time.

➢ Rephrased the second statement/paragraph to clarify: "After desorption, sample loop temperature drops in an exponential decay shaped curve due to the decreasing temperature difference $\Delta T$ between coldhead and sample loop. After a desorption at TD ≈ 200 °C, sample loop and coldhead temperature reached similar temperatures after approximately 30 s cool-down time (TA = −80 °C). This time increases to about 90 s at −120 °C cold head temperature until $\Delta T$ reaches approximately zero. Considering the total run times shown in Table 1, sample loop cool-down to coldhead temperature is not a limiting factor to the overall cycle time."

➢ Please note: the term "sample loop" was changed to "trap" or "preconcentration trap" in response to comment #12 of referee #2.

**16 -** *Page 7 Line 16 ff: Measurement of the trap temperature and thus its control seems to be difficult with a sensor on the trap surface as it is affected by the cold plate. Would it be possible to use the sensor mounted within the second trap to control the temperature of the first although the traps are not identical?*

➢ While possible in principle, there are two reasons not to do so. First, construction and insulation would have to be equal for both traps, which is hard to ascertain. The other reason is that heating both traps would double the heat flow into the coldhead during desorption and thereby increase total cycle time.

**17 -** *Page 9 section 3: Would a title similar to "performance characteristics" be better?*

➢ Rephrased accordingly.

**18 -** *Page 9 section 3.1: As detailed above to improve readability detailed information on the preconcentration unit (e.g. Stirling cooler, MFC) should be given in section 2 and deleted here. Section 3.1 should mainly contain information on the specific set-up used to determine "performance characteristics" as breakthrough volume etc.*

➢ Please refer to our answer on comment #3.
  o Added a link to table 1 (technical details) at the end of the introductory part of sect. 3.
➢ In our opinion, the few technical details given in sect. 3 of the revised version of the manuscript do not hinder its readability and we therefore suggest to kept them.

**19 -** *Page 10 Line 19: The wording "stripe the air of nitrogen etc." should be rephrased.*

➢ Rephrased accordingly: With this preconcentration technique, the most abundant constituents of the air (nitrogen, oxygen and argon) are mostly removed from the sample.

**20 -** *Page 11 Line 8: In Table 2 the relative response is given in "%" please use consistent terms.*

➢ Changed as suggested from "1" to "100 %".
* * *
**21 -** *Page 13 – Page 14: Different wording are used to differentiate detrimental contamination effects: sample loop memory, system memory and system contamination. Please unify and simplify the wording and the ways how to test them, if possible beginning of the section 3.3.*

➢ We thank the reviewer for pointing out this inconsistency.

➢ Rephrased the introductory paragraph of sect. 3.3: "[…] retention time variance during a measurement series ( section 3.3.1). Additionally, the appearance and quantity of analyte signals in measurements of an analyte-free gas after sample measurements determine the number of analysable substances and ultimately measurement data quality. The discussion of analyte residues can be found in  (section 3.3.2). "

➢ Rephrased the introductory paragraph of sect. 3.3.2: "Analyte residues can represent an inherent system contamination (1) or consti-tute a remainder from the previous sample (memory effect, (2)). Both types of residues can originate from different sources like the adsorptive material (sample loop), valve membranes etc. They are differentiated by either an always-present blank signal (1) or a blank signal that decreases to zero in repeated measurements of an analyte-free zero gas after sample measurements (2)."

➢ P. 14 l. 8 and l. 10: added "or contamination" as a differentiation between memory and contamination is not possible in this context.

➢ P. 14 l.10: Added an explanation on why two methods were used for residue investigation (unloaded injection and zero-gas): "The differentiation between (a) and (b) allows a separate investigation, which memory effect or contamination could potentially be reduced by the choice of adsorptive material or parameters of the desorption process (e.g. $T_D$) (a) and which part has to be attributed to tubing, stream selection etc. (b)"

➢ Please note: the term "sample loop" was changed to "trap" or "preconcentration trap" in response to comment #12 of referee #2.
* * *
**22 -** *Page 14 Line 18: "Concentration in the previous run"*

➢ Rephrased as the reviewer suggested: […], so extremely elevated concentrations of low-volatile substances in the previous run might lead to a memory effect […]
* * *
**23 -** *Page 16 Line 3 ff: It would be good to state first what the inter-comparison is based on: The laboratory based instrumentation (GC-TOFMS) is compared to GC-QPMS and GC-MS. As GC-TOFMS and GC-QPMS are based on a similar preconcentration setup, and the preconcentration setup should be tested, it is unclear what we can learn from that.*

➢ We thank the reviewer for pointing out that the internal instrument comparison has to be differentiated from the comparison to the external instrument data.

➢ Rephrased the introductory sentence to clarify: "To ensure internal consistency of our laboratory instrumentation, five air samples were analysed with the GC-TOFMS instrument (Obersteiner et al., 2016) and compared to our reference GC-QPMS (gas chromatograph coupled to a quadrupole mass spectrometer) which uses a similar preconcentration setup (Hoker et al., 2015)" and further (p.16 l.16) "To test the overall performance, the comparison is extended to include in-situ measurement data from the online monitoring Medusa which uses a completely different preconcentration setup […]".

***24 -*** *Page 17 Line 14: Please give details on the argumentation related to the "potential temperature".*

> ➢ The motivation behind the tracer-tracer correlation is the study of atmospheric transport in the upper troposphere and lower stratosphere. Transport processes in the atmosphere are mostly adiabatic. Moving an air parcel along a surface of equal potential temperature (isentrope) does not require a change of its internal energy. In the stratosphere with its stable stratification, the quasi-horizontal transport along the isentropes is therefore fast while exchange between isentropes is slow. When considering the tracer-tracer correlation shown, a point in the correlation therefore represents air with a specific "transport history". As atmospheric transport is not the focus of this manuscript, we did not include any more information on why we use potential temperature to generate the colour code and not e.g. height where the measurement took place. Leaving out this information furthermore does not impede the conclusion that such a compact correlation can only be observed if measurement precision is sufficient.

**Comments of B. R. Miller, referee #2**

*Some general comments: This manuscript describes the relatively simple design of a custom-designed preconcentration unit, which employs a Stirling cooler as the cold source for cryogenic preconcentration of trace atmospheric compounds. The basic design is common to several of their instruments, results of which have been published recently. The lack of need for an insulating vacuum chamber for these cold parts is an appealing characteristic as it greatly simplifies design, reduces cost and weight. The rapid analysis time that they have achieved also makes this a desirable field and laboratory instrument. The focus of the current manuscript is a detailed description and characterization common to these instruments, going beyond what was published in Hoker et al., 2015, Obersteiner et al., 2016 and Sala et al., 2014. The authors do present in-depth results of the chromatographic peak shape resulting from cryofocus and subsequent desorption of analytes. They also discuss results from analyte breakthrough tests and report some of their findings of system contamination.*

*I appreciated the depth of investigation of the 'analyte residues' issue. In such preconcentration methodologies, and at such low analyte mole fractions, these contamination issues can be a very difficult problem to investigate and rectify. So by sharing their observations, the authors are contributing to a knowledge base that may someday be useful in developing some remedies.*

*1 - To add to the value of this residue investigation, I would like to see them include, for example in Table 4, which analyte contaminations they attribute to 'carry-over' from a previous sample, versus those derived from system hardware (e.g., outgassing of polymers, or desorption from surfaces), versus those derived from the carrier gas. In the text, they do allude to such finding with regard to a select set of analytes, but stop short of specification for each of the dozens of compounds listed in the table. Undoubtedly, there will be some analytes for which attribution is not readily discernible, but I would hope that they are able to elaborate with more useful details regarding more clear-cut cases.*

➢ We agree with the reviewer that some value can be added to the discussion with a more differentiated view on analyte residues. We therefore added "*m*" for residues we consider to be carry-overs ("memory" to keep the terminology used in the manuscript) and "*c*" for contaminations in table 4 (now table 5; in brackets, added to columns 5 & 6). Please note that we cannot give a clear source attribution of contaminations (or also memory) for each substance analysed in the frame of this work. Speculations with regard to contamination sources would not be of much use in our opinion as potential sources might be different for different instrumental setups, individual sources might disappear over time ("aging") etc.

➢ Supplement p.1, Table 5 Description. Added information to clarify the implications of the displayed results; (l.12): "Residues that a constant background (contamination), are marked with a "c", ones that represent a memory effect from a preceding sample are marked with an "m"".

➢ Please also consider our response on comment #16, which also refers to contamination source attribution.

***2 -*** *Results from Table 4 present some difficulty in interpretation. For example, iodomethane shows a HayeSep D residue of 43.9%. This is, as defined in the caption, the ratio of the blank gas response to the preceding 1 L air sample, expressed as a percentage. The blank gas injection is defined as an injection of purified helium. From this information, we cannot deduce whether this high percentage is attributable to carry-over of analyte from the air injection into the subsequent blank injection, or if the carrier gas is contaminated with an iodomethane-like substance or if some hardware upstream from the adsorbent tube is outgassing or desorbing the contamination.*

➢ The reviewer is correct in criticising that no differentiation between carry-over and contaminations is made in table 5 (former table 4). This has now been changed; please see our response to comment #1.

➢ Regarding the iodomethane residues, please also see our response on comment #17. This substance is indeed difficult to analyse, although repeated measurements of the same reference gas show decent repeatability. We observe high residues in zero gas measurements with all three instruments here in Frankfurt and also with the GhOST-MS during the PGS campaign. We consider them to be both contamination and carry-over, as we found residues in the unloaded trap injections as well as a constant background signal in the helium we use as carrier gas and zero gas.

*3 - Further difficulty in interpreting table 4 stems from the precision limitation that they impose. The authors exclude analytes from Table 4 that "...show poor precision 10%." With a 43.9% residue of unknown origin, how is a <10% precision justified for including iodomethane in this table? If this contamination comes from the carrier gas, and therefore might be assumed constant, one could correct responses accordingly, and obtain reasonable precisions and accuracies despite contamination. But carryover requires a very different data work-up (e.g., modelling the carry-over). Some explanation is in order here. There are other examples, e.g., HFO-1234yf, chloroethane, tetrachloroethene, etc.*

 ➢ Regarding the 10 % precision limit and some very high residues, please refer to our reply on comment #2 and #16. Note that the 10 % limitation means precision, not accuracy. Considering the large analyte residue of more than 40 % for CH$_3$I (both contamination and carry-over), we would not claim to be able to measure CH$_3$I without further data processing.

 ➢ System contaminations are also not that easy to characterise and correct in our opinion as they might depend on variable parameters like the age of the stainless steel tubing used (e.g. COS and CH$_3$Cl contamination), membrane aging, sample moisture etc. Carry-over is potentially more reproducible and can be modelled after investigation by dedicated experiments. With the manuscript, we did however not intend to provide a viable correction for any of the encountered contamination and carry-over issues.

 ➢ To address this in the manuscript, we rephrased the last paragraph of the summary section, p.19, l.32: "Relatively large amounts of hydrocarbons remained in blank measurements. These  residues are not an inherent problem of the preconcentration setup but more likely due to the adsorptive materials, carrier gas or valve membrane materials used. We do not attempt to present a viable correction method for any of the encountered analyte residues here. More dedicated experiments are necessary to account for analyte-specific memory effect and/or contaminations e.g. by modelling the carry-over from one sample to another and subtracting contamination background. By doing so, the applicability of the preconcentration unit can potentially be extended to quantitative hydrocarbon analysis".

***4 -*** *In reading the manuscript, it seemed odd to me that they omitted detailed analysis long-term reproducibility results, which for me is one of the principle 'bottom-line' results of characterization. The sharp, Gaussian-shaped peaks that they obtain are indeed a desirable trait of such an instrument. But in the intended applications of such an instrument, the value in the measurements comes from short-term repeatability and/or long-term reproducibility.*

> ➢ It is true that the manuscript leaves out instrument characterisation results regarding e.g. measurement precision or reproducibility. These characteristics are however not exclusively influenced by the preconcentration system but also by chromatography and detection. The influence of individual components is often hard or even impossible to disentangle. Details on such characteristics of our laboratory systems were given in the respective publications, Sala et al. (2014), Hoker et al. (2015) and Obersteiner et al. (2016). After the recent PGS campaign, we recently conducted characterisation experiments with the GhOST-MS, so up-to-date details for that instrument will follow in the future. We consider a discussion of "overall" instrument characteristics as beyond the scope of this work as it moves the focus away from the preconcentration system.

> ➢ To let the reader know of this exclusion, we added a sentence in the introduction, p. 3, l. 26: "Only characteristics of the preconcentration setup are discussed; instrument characteristics such as e.g. measurement precision or reproducibility can be found in the respective publications." and at the end of the introductory paragraph of sect. 3: "Please refer to the respective publications for the discussion of instrument characteristics like e.g. measurement precision or reproducibility which are not exclusively related to the preconcentration setup.".

***5 -*** *For a repeatability example, in their aircraft application, what uncertainties might one expect in a vertical profile so that an observed altitude gradient was statistically discernible, or not?*

> ➢ Added information on measurement precision and uncertainty to the caption of Fig. 8: "Preliminary measurement precision and calibration uncertainty: 0.4 % / 1.7 % (Halon 1301), 0.2 % / 0.9 % (Halon 1211)".

> ➢ The compactness of a correlation in the atmosphere can strongly vary depending on e.g. transport and mixing histories of the sampled air parcel; it is therefore hard to give an estimate for the true atmospheric correlation. The observed level of compactness can only be as high as measurement precision allows – which is why we choose this example to demonstrate measurement precision in the field. However, it is not the scope of this manuscript to analyse tracer correlations or atmospheric transport.

**6 -** *In a long-term monitoring application with flasks (i.e., their laboratory instrument), they present a favourable comparison with the AGAGE Medusa instrument at Mace Head, Ireland, for one compound, CFC-12 over one summer. But for characterizing long-term reproducibility suitable for such monitoring programs, one needs to demonstrate this on time-scales of a year or more to argue the potential for comparability and compatibility of datasets. The one brief comparison example falls short of proving the "highly accurate measurement results" that they claim, and it would be beneficial to see reproducibility results for all compounds for which they believe their instrument is suitable to measure.*

➤ We agree with the reviewer that five data points of one substance are insufficient to attest comparable long-term monitoring results. This is also not the focus of this work. With the example shown, we want to demonstrate that good agreement of results from different instruments that both use the described preconcentration system and the Medusa-GC-MS can be achieved, at least for substances that are not subject to other issues like system contamination etc. For this reason, we would prefer not to extend this intercomparison in this paper.

➤ We took part in the InGOS Halocarbon Round Robin Intercomparison (IHRRI) with both our GC-MS instruments, so more results on the accuracy of our measurements will be found there, hopefully in the near future. Furthermore, we are working on the evaluation of the MHD time series that we analyse routinely with our GC-QPMS. Hopefully, we can prepare results including a comparison the Medusa-GC-MS at MHD station for a future publication.

*7 - Another component of characterization that the authors appeared to have overlooked is the potential for CO2-induced artefacts. The methodology appears to offer little in terms of mitigating the presence of this atmospheric constituent that is a million times more abundant than their target analytes. They do make the point that they do not attempt analysis of analytes which elute under the CO2. On their 20 mg of adsorbent, chilled to -80 C or -120 C, they need to show that the co-adsorption of CO2 does not adversely affect adsorption and/or desorption of their analytes. Consider that a 1-L sample of ambient air has order 0.4 ml STP of pure CO2, enough gas to fill a 50 cm length of their 1 mm I.D. tubing. While it is unlikely that they have quantitative adsorption of all the CO2, we aren't given any evidence that sufficient CO2 does not remain to disrupt the desorption of the other analytes and/or disrupt the sensitivity of the detector. In instruments in which no CO2 mitigation is performed, I have observed the co-desorption of CO2 to cause peak width and shape changes in certain co-adsorbed analytes. I have also observed short-term MS detector response reduction due to injection of CO2-containing samples, with subsequent 'rebound' in sensitivity with CO2-free samples. An experiment that the authors could perform to examine this detector issue is to sequence injections of CO2-containing and CO2-free air-like mixtures mixed in with injections of a reference standard gas. If the amount of injected CO2 is having an effect on the reference gas sensitivity, it will be apparent when the sample preceding the reference contained or did not contain CO2.*

- ➢ The reviewer is correct in suggesting that there could be $CO_2$ induced artefacts.
- ➢ We added a sentence on p.19, l26 to acknowledge this fact; "Depending on GC- and detection system, this could induce artefacts in the detected signals […]".
- ➢ … and extended the discussion in sect. 3.2: "[…] like CO2 are however trapped, depending on adsorption temperature. Elution of such species from the GC column after thermodesorption and injection can cause problems with regard to chromatography (e.g. peak tailing) as well as detection (e.g. detector saturation), depending on GC configuration and detection technique. With the setup described here, the elution of CO2 limits the analysable substance range as the detector shows saturation during the elution of CO2 (ionisation switched off until tolerable CO2 levels are reached). A CO2 removal technique could therefore significantly improve chromatographic performance and extend the substance range of the current preconcentration system. At lower adsorption temperatures, even with CO2 removal, a similar problem could however be caused by other gases, like e.g. xenon (boiling point: −108 °C), which is still more abundant by three orders of magnitude in the atmosphere than the targeted analytes discussed here".

- We agree that $CO_2$ is probably not trapped quantitatively and it actually doesn't elute in the form of a clearly defined peak that one would expect from a proper desorption. The m/Q 44 ($CO_2^+$) signal baseline drops until about 10 minutes (right flank of the $CO_2$ "peak") which implies that the GC column is heavily overloaded with $CO_2$. For example, the m/Q 44 baseline intensity at the retention time of CFC-12 exceeds the peak apex intensity of the CFC-12 quantifier ion signal on m/Q 85. This could potentially be another possible explanation for the observed peak tailing of other analytes. Besides $CO_2$, also $H_2O$ could potentially cause artefacts, as the $Mg(ClO_4)_2$ sample dryer only allows drying to a few ppm, which is still sufficient to overload the detector and cause difficulties in GC.

- Regarding detector sensitivity, the discussion is specific for the analytical setup and not really adds to a characterization of the preconcentration itself which is why we excluded it in the manuscript. Currently, we account for the issue by switching the MS filament on not until the amount of $CO_2$ eluting from the GC has reached a tolerable level (detector specific). Especially the TOFs suffer from that issue as they do not allow filtering ions of specific m/Q signals but only broad m/Q intervals (high pass filter) before they are extracted towards the detector. The large amounts of $CO_2$ (fragment) ions can, if not filtered or at least attenuated, harm the detector multichannel plate and overload the analogue to digital converter if the filament is switched on too early or its emission current is too high (or if filter settings are not properly set; which is also true for the QPMS). If on the other hand a high pass filter is used, any m/Q < 44 is also suppressed and respective signals are excluded from evaluation. Potential detector sensitivity drift during a measurement series induced by the repeated injections of large amounts of $CO_2$ can be compensated by repeated measurements of the calibration gas within the measurement series (see also Obersteiner et al. 2016). A rebound effect (increasing detector sensitivity) is unlikely in our opinion as ambient air samples always contain $CO_2$ and normally no $CO_2$-free zero gas is measured within a measurement series.

*8 - Table 2 and corresponding text present unbelievably precise chromatographic retention times, given that this represents 112 injections of various samples. Are these the correct units? Taking CFC-11 for example, 0.006%/100\*7.25s\*60s/min yields 0.19s standard deviation for the CFC-11 retention time. Also, if the fourth column truly is a variance, then the units would be the square of the measurement units, i.e., sec(sup)2. Same applies to line 25 of page 13, where the units would be sec(sup)2.*

➢ We thank the reviewer for pointing out that there is missing information here. The values shown were mean values over three measurement series. We extended the data basis by another measurement series and rephrased the table caption to clarify: "Values derived as arithmetic means over 4 measurement series from different dates (April 2015 to June 2016), comprising 149 individual measurements (~37 per series) of 19 different ambient air samples using the ramped GC program".

➢ We checked the calculation and found it to be correct. The unit of the variance shown in table 4 is changed to $s^2$. We also added the information on p.13, l.25: "Four measurement series were investigated, comprising 149 individual measurement and 19 different ambient air samples".

➢ To show that there is actually more variability between measurement series than within individual measurement series, we added the min-to-max range of the mean retention time variance.

***9 -*** *Retention times vary over time scales on the order of a week in such instruments due to several factors. First, their retention times should vary inversely with ambient pressure. Their electronic pressure control (EPC) of their gas chromatograph references atmospheric pressure such that as ambient pressure increases, the chromatographic pressure gradient between EPC and the absolute vacuum of the detector increases, causing retention time to decrease.*

> ➢ We thank the reviewer for providing this information to the discussion, although we think that this aspect is beyond the scope of this manuscript as it refers only to the GC system. We therefore did not investigate the dependence of retention times on ambient pressure. The basic question here would be against what the EPCs are referenced (ambient air pressure, vacuum etc.).

***10 -*** *Another run-to-run variable is the flow resistance of the gas in the heated adsorbent tube. Figure 3 illustrates quite large temperature oscillations during desorption. Are these oscillations really so reproducible that the gas viscosity is identical in each run?*

> ➢ The oscillation is a result of the heater's pulse width modulation – which had a quite long period of 100 ms and 1 ms minimum increment when the results shown in Fig.3 were recorded (see also Fig.3 caption). We recently improved that to 10 ms period and 20 μs minimum increment (see description in sect. 3.1) but found no evident improvements in injection repeatability (.1 kHz is still a relatively low PWM frequency). This suggests that the temperature oscillation plays a minor role for the actual trap (resp. desorption process; at least in terms of overall repeatability / measurement precision), although it is observable in the dummy trap. The more important factor is presumably the repeatability of the short-time mean temperature of the trap during desorption.

> ➢ We added two sentences on the issue to the manuscript, p.8 l.10: "An effect of this temperature oscillation during desorption on gas flow through the adsorbent (and thereby on injection) cannot be excluded. However, our experience with different heater setups (feedback controlled and deterministic, with different pulse width modulation periods) suggests that it plays a minor role at most for the actual trap; at least in terms of overall measurement repeatability".

***11 -*** *The authors are commended for making the substance breakthrough experiments. However, their conclusions regarding "quantitative trapping" (e.g., page 11, line 8) are only valid if one assumes that the instrument is linear in response, an assumption for which no evidence is provided. For example, the greater than 100% response ratio of ethyne may be attributable to "memory effect" as they claim, but the 0.3% to 1.4% 'residue' for ethyne listed in table 4 does not support the observed 109% 'breakthrough result' reported in table 2. There are more definitive tests for characterizing breakthrough, several of which are described in the 1999 thesis work of Dr. Brian Greally from the University of Bristol, UK. Use of the technique of varying the volume of sample gas is one method to examine system non-linearities, but this must be accompanied by a confirming technique such as analysis of a suite of air-like gases of differing analyte mole fraction.*

> ➢ We thank the reviewer for pointing out that the results of the volume variation experiment require a more differentiated interpretation.

> ➢ Please consider that it is beyond the scope of this work to investigate detector non-linearities; for that matter, refer to e.g. Hoker et al. 2015 or Obersteiner et al. 2016. Results from volume variation experiment also suffer from the fact that not the concentration of individual analytes is varied but that of all. This might cause a different detector response than one would achieve with a single elevated concentration and might also vary depending on the used reference gas as air matrices could be different. Consequently, also in the investigation of non-linearities, the volume variation experiment has to be interpreted against this background. The other experiment, a series of analyte dilutions in e.g. synthetic air, on the other hand heavily relies on the quality of the dilution, i.e. how accurately dilution factors can be determined. These might furthermore be substance-specific (depending on e.g. volatility). Unfortunately, we do not own such a dilution series that contains high mole fractions needed for a breakthrough experiment in Frankfurt.

> ➢ Regarding the 109 % relative response of ethyne for the high preconcentration volume, a detector non-linearity cannot be totally excluded. However, the message here is that a breakthrough is unlikely due to the relative response greater than 100 %. We changed the wording of the respective sentence (p. l.) to clarify: "ethyne was also analysed on the Unibeads 1S trap which gave a quite different result with a deviation from linear response of +10.1 % ± 0.51 %, thus breakthrough is unlikely. The positive, non-linear response is caused potentially by a system blank (see also section 3.3) or non-linear detector response". Moreover, we think that it is difficult to compare a zero-gas blank value with 1 L preconcentration volume with an overestimated concentration in a volume variation experiment with a preconcentration volume of 10 L.
* * *
*Specific comments:*

**12 -** *The authors refer to their micro-packed adsorbent tube as a 'sample loop', which I find an unusual use of the term. The use of the term 'sample loop' in the general literature usually refers to an open tubular device which is used to measure the sample volume, typically by pressure measurement under fixed temperature and volume assumptions. I would suggest using a more common terminology for their adsorbent tube.*

➢ The term is indeed unappropriated; changed to "preconcentration trap" or "trap" (several places in the manuscript, including Fig. 1 and Fig. 2).
* * *
**13 -** *Pg. 3, line 9: LN2 and Ar used for cooling in these kinds of instruments are not typical examples of 'refrigerants', but of cryogens. Refrigerants undergo expansion (cooling) and subsequent recompression cycles.*

➢ We thank the reviewer for commenting on this imprecise wording.
➢ Changed title of the manuscript to "A versatile, refrigerant and cryogen-free cryofocusing thermodesorption unit for preconcentration of traces gases in air"
➢ Rephrased wording in the abstract: "Reliable operation is ensured by the Stirling cooler, which does not require refilling of a liquid refrigerant neither contain a liquid refrigerant nor requires refilling of a cryogen. At the same time it while allowing allows significantly lower adsorption temperatures compared to commonly used Peltier elements".
➢ Introduction, p.3: replaced "refrigerant-based" with "cryogen-based" as suggested.
➢ Summary, p.19, l.2: added "cryogen-based": "A single-stage, refrigerant- and cryogen-free sample preconcentration unit".
➢ P. 19, l.18, changed: "refrigerantcryogen-based" as $LN_2$ / LAr based cooling options are meant here.

**14 -** *Pg. 4, line 28: "self-written control software" should be changed to "custom-written" or similar.*

> ➢ Rephrased as suggested, "custom-written"

**15 -** *Pg. 10, line 19: ": : :strips the air of its most abundant constituents: : :" makes it sound like you are preconcentrating N2, O2, etc., and not the trace target compounds.*

> ➢ Rephrased, please see our response on comment #19 of referee #1.

**16 -** *Pg. 14, line 16: "In general, most of the detected analyte residues are most probably caused by system contaminations (HFCs from fittings, solenoid valve membrane etc.) : : :" By 'solenoid valve membranes, are you referring to those in the EPCs?*

> ➢ Yes, the EPCs of the GC might be a source; the helium that we use as blank/zero gas also goes through an EPC so that we get a suitable pressure for preconcentration. We think that basically anything that is not full metal could be a source, from back pressure regulator membranes to the coating of the rotary valve head. It is therefore hard to say which source is dominant or which source emits what compound. To emphasise this aspect, we added information and rephrased beginning on p.14, l.17: "A distinct attribution of specific sources was not attempted. Please also note that potential contamination sources might be different for different instrumental setups, individual sources might disappear over time ("aging") etc. Regarding system memory (including the trap), the amount […]".

*17 -* *Table 4 caption: Why is it that some analytes that show large residues, e.g., iodomethane at 43.9% for HayeSep D, are included in this table when the caption states that those that ": : :show poor precision 10% were excluded?*

➢ The values reported in table 5 (manuscript's supplement) represent relative analyte signals as is explained in the description text. In the analysis of repeated measurements of the same sample, iodomethane showed a quite stable detector response, thus the measurement precision we deduced is relatively good: approximately 3 % for both in-situ as well as laboratory instruments (current, unpublished value; the value named in Hoker et al. 2015 is a bit better (~1 %)). Considering the high residues, we have to conclude that at the moment we can measure $CH_3I$ with decent precision but likely inaccurate.

➢ Values for measurement precision are not named in this manuscript; please also see our response on comment #4.

*18 -* *Figure 4 caption and elsewhere: "n-propane" should be simply "propane"*

➢ Rephrased accordingly.

**Comments of referee #3**

*In this article Obersteiner and co-workers present a novel device to sample highly volatile compounds without the need of a cooling agent. Although it has been designed for two specific gc-ms systems, its use in many applications, where cooling agents are not available, is conceivable. The device has been very thoroughly characterized and its strengths and weaknesses are described in the paper. I only have some remarks to improve the structure and the readability of the publication.*

*I would suggest publishing the article with minor revisions in Atmospheric Measurement Techniques.*

*1 - Page 2, Line 13: The term 'direct measurement' is not unambiguous., I would rephrase this sentence.*

➢ Rephrased the sentence as suggested: "[…]too low for immediate detection and quantification by means of instrumental analytics without further sample processing steps"

*2 - Page 3, Line 3 Is there literature on the PerkinElmer system?*

➢ The PerkinElmer "Turbomatrix" thermodesorber can be found in publications (e.g. Palmer et al. (2005), doi:10.1071/EN05078, iodocarbon analysis or Jones et al. (2009), doi:10.5194/acp-9-8757-2009, also iodocarbon analysis); however not very frequently in publications that evolve around the analysis of halogenated trace gases to our knowledge.

*3 - Page 4, Line 19 Write flushed in direction opposite to sampling flow (instead of backwards)*

➢ Rephrased as suggested.

*4 - Page 5, Line 30 The term chromatographic runtime is not clear to me. Does it mean that the chromatographic separation is the time limiting step?*

➢ The reviewer is correct in pointing out that this expression lacks definition; rephrased to clarify: "the total duration of the chromatogram (chromatographic runtime)"

*5 - Page 6, Line 10 Use time resolution (instead of number of measurements per time)*

➢ Rephrased as suggested.

**6 -** *Page 6, Line 16 reach (instead of reached)*

➢ Rephrased as suggested. Please also see our changes in this paragraph in response to reviewer #1, comment #15.
* * *
**7 -** *Page 6, Line 11 Why should an increased preconcentration flow increase the time resolution? Is it due to the decreased size of the cold head?*

➢ We thank the referee for pointing out that there is something unclear. We rephrased and complemented on p.6, l.3: "[…]various factors determine the minimum cycle time (i.e. sample measurement frequency) including:
  - ○ Sample preconcentration: volume of the sample to preconcentrate and preconcentration flow
  - ○ Sample desorption: duration and $T_D$ as well as insulation of the trap
  - ○ Cool-down of trap and coldhead after desorption: targeted adsorption temperature $T_A$, cooling capacity (i.e. heat lift around $T_A$) and coldhead insulation as well as ambient temperature

  Shortening of any of these steps can theoretically shorten the overall cycle time and thereby increase time resolution; however, there might be no benefit in doing so if there are other limitations like the time it takes to record the chromatogram of a sample injection.

➢ Added information on p.6, l.7: "However, the overall time resolution of the laboratory instrument is limited by the GC with a total time of 19.6 minutes per chromatogram".

*8 - Page 6, Line 18 The difference of total run time, overall cycle times and minimum cycle time is not clear to me. Is the time resolution of e.g. the GhOST-MS equivalent the minimum cycle time plus the time for cool-down, or is the cool-down phase already during the GC runtime? You write that the sample-loop cool down time is not a limiting factor. But on page 17 line 8 you write that the cycle time of the GhOST-MS is indeed limited by the cooldown of the adsorptive material. Isn't it the cooldown time? Maybe a schematic diagram of the runtime with the cool down-time, the desorption time and the chromatographic runtime could help to avoid any confusion on the different cycle times.*

➢ We thank the reviewer for pointing out that there is missing information here, i.e. the total time it takes to measure one sample, including preconcentration and gas chromatography (again including GC temperature program and cool-down of the column to starting temperature; see also our reply on comment #7).

➢ Added a column (5) to table 2, "Experimental total time needed for one measurement". Rephrased header of column 4 to "minimum preconcentration cycle time" to clarify the difference. Please note: an additional table was introduced in response to reviewer #1; the "cycle-time table" is now table 2.

➢ The GhOST-MS uses a low thermal mass (LTM) module that contains the main chromatographic column and allows very short chromatograms. The chromatographic runtime of the GhOST-MS is 2.9 minutes plus approx. 1.2 minutes needed to cool the column down to starting temperature after a chromatogram has finished (ambient air driven by a 150 W fan). The total measurement time in this case is actually limited by both the GC column cool-down time and the cool-down time of the coldhead (comp. to our other instruments, only $T_A \sim -72$ °C can be achieved).

*9 - Page 7, Line 27 A short description of the 'stages' should be already included here.*

➢ The wording of the named section is indeed slightly inconsistent; rephrased the sentence beginning in l. 26, p. 7: "Very good results were achieved with a two-stage, deterministic heater setup with a fast heat-up (stage 1), a small overshoot  of $T_D$ and preservation of $T_D$ (stage 2) with only a small drift and fluctuation."

**10 -** *Page 10, Lines 7-14 The importance of desorption for the chromatographic peak shapes is very nicely discussed in chapter 3.3. Hence, Figure 4 should be discussed in chapter 3.3 and not already here.*

➢ The reviewer is right that Fig. 4 better fits in the discussion of desorption than in the introductory paragraph where is just shown and not discussed.

➢ Moved the reference to Fig. 4 to sect. 3.3, p. 13, at the end of the introductory paragraph (slightly rephrased): "Figure 4 shows a typical chromatogram recorded after desorption and injection of a preconcentrated ambient air sample for three selected mass to charge ratios (m/Q).".

➢ Added another reference to Fig. 4 on p.13 l.22 : "[…] as these are only very little influenced by the chromatographic system (see also Figure 4)".

**11 -** *Page 10 Line 20 When atmospheric ozone is trapped in the cryofocus it can degrade the alkenes mentioned in the supplement. Has the influence of ozone on the recovery of VOCs been investigated?*

➢ This is an interesting aspect but has not been investigated. Considering that ozone destruction is catalysed on hot stainless steel surfaces, we would assume that the amount of ozone that reaches the adsorptive material is probably small. This is however speculation and was not tested experimentally. Our focus is on the analysis of halogenated tracers; results for some VOCs were included in this work to show that there is potential to extend the substance range by at least some species of this class. A detailed investigation would however go beyond the scope of this work.

➢ To acknowledge this fact, we added a sentence to the discussion of adsorption, sect. 3.2, first paragraph: "Interactions of other, reactive species like ozone with analytes (e.g. alkenes) during trapping and desorption were not investigated in this work".

***12 -*** *Page 11, Line 1 How is the back pressure of 2.5bar generated? From figure 1 the sample loop should have ambient or reduced pressure during sampling.*

➢ The reviewer is right in pointing out that it is not made clear in which part of the system the pressure applies. The value refers to the value indicated by the low pressure manometer of the back pressure regulator of the sample flask which was used for the test (high pressure sample, ~145 bar abs.).

➢ Added explanation in the text to clarify: "[…] sample back pressure of 2.5 bar abs. (back pressure indicated by the regulator of the sample flask)".

***13 -*** *Page 13, Line 14 Peak tailing is by definition on the right flank, so I would propose to simply write peak tailing or tailing.*

➢ Rephrased accordingly, also on p. 15.

***14 -*** *Page 13, Line 30 You write that the tailing effect could potentially be reduced by refocusing the high-volatile analyte fraction on a second sample loop. How can this be achieved? Do you need a third valve for it, or would you place it in front of the main column?*

➢ By the refocusation, the spatial spread of the analyte molecules on the adsorptive material / GC column is reduced. There are different ways how to achieve refocusation; one of them would be to cool the first part of the column (pre-column in our setup) or even the whole GC oven e.g. with $LN_2$. There are commercial solutions for this method available from GC manufacturers or suppliers of analytical hardware like e.g. Gerstel, Germany. The other method would be to add a second sample loop after the first one, which is cooled and heated separately. A possible setup is realised in the Medusa preconcentration system, see Fig. 1 in Ben Miller's paper on the Medusa (Miller et al. 2008).

➢ Rephrased on p.13 l.29: " Parts of this tailing effect which originate from desorption could potentially be reduced by refocusing the high-volatile analyte fraction on a second trap (e.g. Miller et al., 2008)".

➢ Peak tailing could also (partly) be caused by tubing and valve used to connect pre- and main column in our GC configuration, so the adsorption/desorption process is potentially not the only origin of this effect. Another factor could be $CO_2$ loading of the GC column, please see our reply on comment #7 of reviewer #2. This was however not tested experimentally so we did not include this speculation in the manuscript.

**Further changes to the manuscript**

➢ Multiple occasions: replaced "manuscript" by "paper"

➢ Abstract, l.18: "" (superfluous)

➢ p.2 l.26: rephrased for better definition: "[…] and preferentially cryogen- and refrigerant-free, pure electrical operation. Liquid cooling agents (cryogens) like […]".

➢ p.19 l.4: rephrased for more precise wording: "trap and desorb a wide range of halo-halogenated trace gases and potentially also hydrocarbons".

➢ p.19 l.9: replaced: "analyte residues".

➢ Acknowledgements: added "Finally, we thank B. R. Miller and two anonymous referees for reviewing the manuscript".

➢ Supplement p.1, Table 5 Description. Rephrased, l.12: "Substances that are not detected regularly in ambient air samples or show poor measurement precision ≥ 10 % were excluded from the analysis" to clarify.

**Mark-up version of revised the manuscript**

- see next page -

**A versatile, refrigerant- and cryogen-free cryofocusing-thermodesorption unit for preconcentration of traces gases in air**

F. Obersteiner[1], H. Bönisch[2], T. Keber[1], S. O'Doherty[3] and A. Engel[1]

[1] Institute for Atmospheric and Environmental Science, Goethe University Frankfurt, Frankfurt, Germany

[2] Institute of Meteorology and Climate Research, KIT, Karlsruhe, Germany

[3] School of Chemistry, University of Bristol, Bristol, United Kingdom

*Correspondence to*: F. Obersteiner, obersteiner@iau.uni-frankfurt.de

**Abstract.** We present a compact and versatile cryofocusing-thermodesorption unit, which we developed for quantitative analysis of halogenated trace gases in ambient air. Possible applications include aircraft-based in-situ measurements, in-situ monitoring and laboratory operation for the preconcentration of analytes from flask samples. Analytes are trapped on adsorptive material cooled by a Stirling cooler to low temperatures (e.g. −80 °C) and desorbed subsequently by rapid heating of the adsorptive material (e.g. +200 °C). The setup neither involves exchange of adsorption tubes nor any further condensation or refocusation steps. No moving parts are used that would require vacuum insulation. This allows a simple and robust single stage design. Reliable operation is ensured by the Stirling cooler, which does not require refilling of a liquid refrigerantneither contain a liquid refrigerant nor requires refilling of a cryogen. At the same time it, while allowingallows significantly lower adsorption temperatures compared to commonly used Peltier elements. We use gas chromatography - mass spectrometry for separation and detection of the preconcentrated analytes after splitless injection. A substance boiling point range of approximately −80 °C to +150 °C and a substance mixing ratio range of less than 1 ppt (pmol mol$^{-1}$) to more than 500 ppt in preconcentrated sample volumes of 0.1 to 10 L of ambient air is covered, depending on the application and its analytical demands. We present the instrumental design of the preconcentration unit and demonstrate capabilities and performance through the examination of analyte breakthrough during adsorption, repeatability of desorption and analyte residues in blank tests. injection quality, analyte breakthrough and analyte residues in blank tests. Application examples are given by the analysis of flask samples collected at Mace Head Atmospheric Research Station in Ireland using our laboratory GC-TOFMS instrument and by data obtained during a research flight with our in-situ aircraft instrument GhOST-MS.

**1  Introduction**

[revised manuscript text omitted]

sample measurement frequency) including:

- Sample preconcentration: volume of the sample to preconcentrate and preconcentration flow
- Sample desorption: duration and $T_D$ as well as insulation of the trap
-  targeted adsorption temperature $T_A$
-  cooling capacity (i.e. heat lift around $T_A$) and coldhead insulation as well as ambient temperature
-
-

Shortening of any of these steps can theoretically shorten the overall cycle time and thereby increase time resolution; however, there might be no benefit in doing so if there are other limitations like the time it takes to record the chromatogram of a sample injection. To give a practical example,

**Table 2** shows cycle times derived from routine operation data. A total time per  preconcentration cycle of 18.6 minutes is necessary if $T_A = -120\ °C$ and $T_D \approx 200\ °C$ is desired – mainly determined by the time needed to compensate the warm-up of the coldhead during desorption. This minimum time interval significantly shortens to 8.5 minutes if $T_A$ is increased to $-80\ °C$ (same $T_D$). However, the overall time resolution of the laboratory instrument is limited by the GC with a total time of 19.6 minutes per chromatogram. Data from the in-situ setup shown in

**Table 2** demonstrates that very short cycle times of 4.1 minutes are possible with a decreased preconcentration volume (100 mL instead of 500 mL; requiring a detector that is sensitive enough),  a slightly higher $T_A$ (~ −72 °C) and a faster GC. General measures to increase time resolution would be to increase the preconcentration flow, reduce the sample size (see in-situ setup), improve the coldhead and trap insulation and increase the cooling capacity.

After desorption,  temperature of the trap drops in an exponential decay shaped curve due to the decreasing temperature difference $\Delta T$ between coldhead and trap. After a desorption at $T_D \approx 200\ °C$, preconcentration trap and coldhead temperature reach similar temperatures after approximately 30 s cool-down time ($T_A = −80\ °C$). This time increases to about 90 s at −120 °C cold head temperature until $\Delta T$ reaches approximately zero. Considering the total run times shown in

**Table 2**, trap cool-down to coldhead temperature  is not a limiting factor to the overall cycle time. Consequently, thermal insulation of the trap could still be increased, thereby decreasing coldhead warm-up during desorption.

**Table 2**, trap cool-down to coldhead temperature  is not a limiting factor to

**2.3 Thermodesorption: preconcentration trapsample loop heater**

Depending on the targeted substance class to analyse and the analytical technique, the requirements for thermodesorption will differ. In case of a gas chromatographic system for analysis of volatile compounds, these requirements are:

- a fast initial increase in temperature to yield a sharp injection of highly volatile analytes onto the GC column,
- no overshooting of a maximum temperature in case of thermally unstable sample compounds or adsorptive material (e.g. HayeSep D, $T_D < 290$ °C)
- preservation of the desorption temperature over a time period for desorption of analytes with higher boiling points
- good overall repeatability, especially of the injection of highly volatile analytes

Desorption heating is implemented by pulsing a direct current (max. 12 V / 40 A, relay: Celduk Okpac; spec. switching frequency 1 kHz, Celduk Relays, France) directly through the sample looptrap tubing which has a resistance of ~0.5 Ω. A temperature sensor (Pt100, 1.5 mm OD) was welded to the outside of the sample looptrap tubing (see also **Figure 2**), for feedback control of the heater temperature. However, mainly due to the thermal mass of the sensor and its proximity to the coldhead (despite the insulation), it was found to give no representative values for temperature inside the sample looptrap during desorption. Differences of around 100 °C were found in comparison to temperature measured within the sample looptrap (equilibrium state; after 2-3 minutes of continuous heating). Nevertheless, the temperature sensor can be (after being characterised) used for feedback control as the indicated values are reproducible. As an alternative to feedback control, a deterministic heater with prescribed output settings can be used. For security reason, measured coldhead and sample looptrap temperature have to be used as heater shutdown triggers in this case.

**Figure 3** shows a comparison of temperature sensor data from in- and outside the empty sample looptrap as well as the coldhead. Very good results were achieved with a two-stage, deterministic heater setup with a fast heat-up (stage 1), a small overshoot between stage 1 and 2 of the heating phaseof $T_D$ and preservation of $T_D$ (stage 2) with only a small drift and fluctuation. With the described heater setup, $T_D$ can be reached within a very short time of approximately 3 seconds. Initial heating rates (first second of heat pulse) were calculated to be more than 200 °C s$^{-1}$ depending on the power output setting. As the sample looptrap is getting warmer, heating rate drops resulting in a mean heating rate of about 80 °C s$^{-1}$ during stage 1.

If a deterministic heater is used instead of a feedback controlled heater,  the tem- perature of the trap becomes directly dependent on coldhead temperature (more precisely:

heat flow from the trap into the coldhead). Consequently, higher output settings are necessary at lower coldhead temperatures to achieve comparable temperatures. On the other hand, if the coldhead gets warmer, trap temperature increases as well. This effect can be observed in **Figure 3** as a slight upward drift of the trap temperature (red curve, temperature measured within the trap) during stage 2. The absolute temperature differences caused by this drift as well as the oscillation amplitude are small (in

**Figure 3**: approximately 20 °C min. to max. and 4 °C standard deviation without trend cor- rection) compared to the temperature difference between coldhead and trap during heating (about 300 °C). An effect of this temperature oscillation during desorption on gas flow through the adsorbent (and thereby on injection) cannot be excluded. However, our ex- perience with different heater setups (feedback controlled and deterministic, with different pulse width modulation periods) suggests that it plays a minor role at most for the actual trap; at least in terms of overall measurement repeatability.

Besides the problem of differing inner and outer temperature of the trap during heating, temperature was not found to be distributed homogeneously alongside the empty trap inside the coldhead. Temperature differences of up to ±30 °C at 200 °C mean temperature were observed with the current setup if measuring temperature at different points within the trap, potentially due to (a) difficulties in accurately measuring the inner temperature (wall contact of sensor) and (b) inhomogeneity in trap insulation as well as variations in tubing wall width or carbon content leading to an inhomogeneous electri- cal resistance and thus an inhomogeneous distribution of heat. These temperature variations might be different or ideally negligible in the ac- tual preconcentration trap. However, the finding underlines the importance of an insulation as homogeneous as possible and suggests that "cold points" (possibility of insufficient desorp- tion) as well as "hot points" (possibility of adsorptive material or analyte decomposition) are possible along the trap, which has to be taken into consideration when setting up and testing the preconcentration setup, i.e. to not exceed the temperature limit of the adsorp- tive material.

**3  **Performance characteristics**

[revised manuscript text omitted]

peak tailing) as well as detection (e.g. detector saturation), depending on GC configuration and detection technique. With the setup described , the elution of $CO_2$ limits the analysa- ble substance range as the detector shows saturation during the elution of $CO_2$ (ionisation switched off until tolerable $CO_2$ levels are reached). A $CO_2$ removal technique could therefore significantly improve chromatographic performance and extend the substance range of the current preconcentration system. At lower adsorption temperatures, even with $CO_2$ removal, a similar problem could however be caused by other gases, like e.g. xenon (boiling point:

−108 °C), which is still more abundant by three orders of magnitude in the atmosphere than the targeted analytes discussed here. Interactions of other, reactive species like ozone with analytes (e.g. alkenes) during trapping and desorption were not investigated in this work.

Regarding preconcentration of targeted analytes, the concept of an adsorption-desorption steady state suggests that at a certain point a breakthrough of analytes occurs, depending on a combination of loading of the solid phase with sample molecules and time to achieve steady state, in turn influenced by sample flow rate and pressure. Consequently, the maximum possible sample volume and/or minimum duration of preconcentration are dependent on the adsorptive material used, volatility (and concentration) of the targeted analytes as well as sample flow rate and pressure. For typical sample volumes of 0.5 L and 1.0 L (at standard temperature and pressure) and a constant sample back pressure of 2.5 bar abs. (back pressure indicated by the regulator of the sample flask), no significant impact of sample preconcentration flow was found within the tested range of 50 mL·min$^{-1}$ to 150 mL·min$^{-1}$ for any of the analysed substances. Higher or lower flow rates and pressure were not possible or suitable for practical reasons like flow restriction and valve operating pressure.

Substance breakthrough (i.e. substance-specific adsorption capacity) was analysed in volume variation experiments, comprising measurements of the same reference air with preconcentration volumes of up to 10 L and referencing the volume-corrected detector response against default preconcentration volumes of e.g. 1 L ("relative response"). Quantitative trapping is then indicated by a relative response of 100 %; a relative response <100 % would indicate an underestimation (i.e. loss by breakthrough), a relative response of >100 % would indicate an overestimation (i.e. increase by a memory effect from the preceding sample). To structure the following discussion, two classes of substances are formed and treated separately: "medium volatile substances" with boiling points > −30 °C (e.g. CFC-12, $CCl_2F_2$) and "highly volatile substances" with boiling points < −30 °C (e.g. HFC-23, $CHF_3$). The substances discussed are selected based on the criteria volatility and (preferably high) concentration. The adsorption of substances with lower volatility (BP > 30 °C) was assumed to be quantitative. Results discussed in the following are displayed in **Table 3**.

***Medium volatile substances.*** As a reference for halocarbon analysis, CFC-12 ($CCl_2F_2$) and CFC-11 ($CCl_3F$) were chosen due to their high mixing ratios of about 525 and 235 pmol·mol$^{-1}$ (ppt, parts per trillion) in present-day, ambient air and moderate volatility with boiling points of −29.8 °C and +23.8 °C. For a volume of 10 L preconcentrated air on the

Unibeads 1S trap, both substances showed a deviation from linear response of +0.6 % ± 0.42 % for CFC-12 and +0.6 % ± 0.22 % respectively for CFC-11. The positive deviation from linearity is still found within the 3-fold measurement precision determined for the experiment and could potentially be an artefact of the detector used which tends to slightly overestimate strong signals and underestimate weak signals; see section 3.4 in Obersteiner et al. (2016). Hence, no significant breakthrough or detector saturation was observed for both substances CFC-12 and CFC-11.

***Highly volatile substances.*** More volatile compared to CFC-12 and CFC-11 but similar in mixing ratio is carbonyl sulfide (COS) with a boiling point of −50.2 °C and an ambient air mixing ratio of typically around 500 ppt. Against 1 L reference sample volume (sample mixing ratio: 525 ppt), COS showed a quantitative adsorption up to 5 L on the Unibeads 1S trap with a deviation from linear response of +0.9 % ± 0.80 %. At 10 L sample volume, a breakthrough occurred giving a deviation from linear response of −35.2 % ± 0.52 %. The substance analysed with highest volatility was HFC-23 with a boiling point of −82.1 °C and a current background air mixing ratio of ~40 ppt. Referenced against a sample volume of 0.5 L, significant breakthrough occurred at a sample volume of 2.5 L with a deviation from linear response of −39.2 % ± 2.75 %. The highest sample volume quantitatively adsorbed in the experiment was 1.0 L with a relative response of −0.3 % ± 2.75 % (HayeSep D trap). A similar behaviour was observed for ethyne ($C_2H_2$), with a sublimation point of −80.2 °C, a mixing ratio of approximately 610 ppt in the sample and a deviation from linear response of −20.2 % ± 1.22 % at 2.5 L sample volume (HayeSep D trap). However, ethyne was also analysed on the Unibeads 1S trap which gave a quite different result with a deviation from linear response of +10.1 % ± 0.51 %, thus breakthrough is unlikely. The positive, non-linear response is caused potentially by a system blank (see also section 3.3) or non-linear detector response. Unfortunately, HFC-23 could not be analysed in ambient air samples for comparison on the Unibeads 1S trap as its ion signals are masked by large amounts of $CO_2$ still eluting from the GC column at the retention time of HFC-23.

Concluding, the adsorption process was found to be substance specific as both HFC-23 and ethyne are comparably volatile but significantly less ethyne broke through despite its 15-fold elevated mixing ratio (Unibeads 1S trap). The comparison of ethyne breakthrough on the HayeSep D and Unibeads 1S trap suggests that the adsorption process is dependent on the chosen adsorptive material. A comparison of adsorptive materials is howev- er not the focus of this work; such a comparative adsorption study was e.g. conducted for me- thane ($CH_4$) preconcentration by Eyer et al. (2014). From the comparison of the breakthrough observed for COS and the quantitative adsorption of CFC-12 and CFC-11, it can be concluded that volatility is the primary factor that determines breakthrough. Quantitative adsorption is not limited by principal adsorption capacity (i.e. the absolute number of molecules adsorbed)

of the adsorptive material and material amount for a sample volume of up to 10 L and an ad- sorption temperature of −80 °C.

**3.3  Desorption**

While adsorption is characterised by the quantitative trapping of highly volatile substances, desorption is characterised by sharpness and repeatability of the injection represented by chromatographic peak shape and retention time variance during a measurement series ( section 3.3.1). Additionally, the appearance and quantity of analyte signals in measurements of an analyte-free gas after sample measurements determine the number of analysable substances and ultimately measurement data quality. The discussion of analyte residues can be found in  section 3.3.2. **Figure 4** shows a typical chromatogram recorded after desorption and injection of a preconcentrated ambient air sample for three selected mass-to-charge ratios (m/Q).

**3.3.1  Peak shape and retention time stability**

To demonstrate injection sharpness, **Figure 5 A** shows the chromatographic signal of CFC-11 eluted from the GC column kept isothermal at 150 °C and **Figure 5 B** the chromatographic signal as observed with the ramped GC program. Both signals generally show a Gaussian peak shape with a slight peak tailing . In comparison, the "unfocused" signal from the isothermal column reflecting the sharpness of the direct injection is wider by a factor of ~3 but still narrow enough to allow for good peak separation in most standard GC methods with runtimes between 10 to 30 minutes; the full peak width at half maximum (FWHM) was calculated to be 6.3 s (0.10 min) for the isothermal peak and 2.0 s (0.03 min) for the focused peak.

Injection quality can further be judged by the stability of retention times of the first chromatographic signals obtained with the ramped GC program, as these are only very little influenced by the chromatographic system (see also **Figure 4**). In particular, there is nearly no refocusing on the chromatographic column. **Table 4** shows retention times and their variability expressed as relative standard deviation and variance as well as the chromatographic signal width (FWHM) of the respective substance. Four measurement series were investigated, comprising 149 individual measurement and 19 different ambient air samples. Variances are less than 0.02 $s^2$ on average. Together with signal width, they decrease reversely proportional to retention time, which shows the increasing influence of chromatographic separation (from HFC-23 to CFC-11 in **Table 4**). Even at incomplete re-focusation by gas chromatography, the desorption procedure of the preconcentration unit gives close to Gaussian peak shapes except a slight tailing .  Parts of this tailing effect which originate from desorption could potentially be reduced by refocusing the high-volatile analyte fraction on a second trap (e.g. Miller et al., 2008). The high repeatability of the injection is shown by the low variability in retention time of the first signals in the chromatogram (**Table 4**).

**3.3.2 Analyte residues**

Analyte residues can represent an inherent system *contamination* (1) or constitute a remainder from the previous sample (*memory effect*, (2)). Both types of residues can originate from different sources like the adsorptive material (preconcentration trap), valve membranes etc. They are differentiated by either an always-present blank signal (1) or a blank signal that decreases to zero in repeated measurements of an analyte-free zero gas after sample measurements (2).

Analyte residues were investigated with (a)  unloaded injections after multiple 1 L ambient air sample injections, i.e. subsequent thermodesorption of the trap without switching to load-position between runs (see **Figure 1**) and (b) the preconcentration of 1 L helium from the carrier gas supply using the same path as the sample, including dryer etc. after multiple 1 L ambient air sample measurements. Analyte residues on the trap (*sample loop memory or contamination*) as well as carrier gas contaminations are investigated by (a) while (b) includes analyte residues within the tubing upstream of the trap, i.e. stream selection, sample dryer etc. (*system memory or contamination*). The differentiation between (a) and (b) allows a separate investigation, which memory effect or contamination could potentially be reduced by the choice of adsorptive material or parameters of the desorption process (e.g. $T_D$) (a) and which part has to be attributed to tubing, stream selection etc. (b).

To get the most complete picture possible, 65 substances were analysed, most of them halo- and hydrocarbons (see supplementary information for a detailed list) on both a HayeSep D as well as a Unibeads 1S trap. Substances with low measurement precision (> 10 %) were excluded from the investigation. In general, most of the detected analyte residues are  probably caused by system contaminations (HFCs from fittings, solenoid valve membranes etc.) or carrier gas contaminations (hydrocarbons) as they show a constant background.

A distinct attribution of specific sources was not attempted. Please also note that potential contamination sources might be different for different instrumental setups, individual sources might disappear over time ("aging") etc. Regarding system memory (including the trap), the amount of a residue is dependent on volatility and concentration, so extremely elevated concentrations of low-volatile substances in the previous run might lead to a memory effect that was not detected in the current investigation with 1 L preconcentration volume of unpolluted ambient air. Detailed results for the two different adsorptive materials tested are discussed in the following.

*Unibeads 1S adsorptive material.* 13 of 65 substances (20 %) did show detectable residues on the trap which did not represent a system memory but a system contamination, e.g. from the carrier gas, sealing materials etc. as they were always present and did not disappear in subsequent unloaded injections. Respective residues were generally larger with increasing boiling point (e.g. propane < benzene). Most of them were hydrocarbons and the halocarbons chloro- and iodomethane ($CH_3Cl$, $CH_3I$) and chloroethane ($C_2H_5Cl$) as well as HFC-134 ($CHF_2CHF_2$). No further CFCs, HCFCs, PFCs or HFCs were detected in the unloaded trap injection (see Obersteiner et al. (2016) for a discussion of detection limits). Of the remaining 52 substances, 36 also did not show any detectable residues in the helium blank. Of the 17 substances that did show residues (contamination and memory effect combined), 7 had residues below 0.5 % of the signal area determined in the preceding ambient air measurement. Again, residues were found mostly for hydrocarbons but not CFCs or HCFCs. Concluding, the Unibeads 1S trap seems to be a good choice for halocarbon monitoring measurements (one measurement per sample) as there were nearly no halocarbon residues in subsequent helium blank measurements.

*HayeSep D adsorptive material.* The HayeSep D trap showed a considerably higher amount of preconcentration trap residues (unloaded injection) with 22 detectable substances from the selected 65 (34 %). Again, most of these substances were hydrocarbons but also some halogenated compounds like Tetrachloromethane ($CCl_4$) and Bromoform ($CHBr_3$). Of the remaining 43 substances, 28 were undetectable in the helium blank (system free of contamination  or memory effect). 13 of the detectable substances showed responses of < 0.5 % relative to the preceding ambient air sample, also including CFC-11 with 0.05 % and CFC-113 with 0.2 %. While the named halogenated compounds $CCl_4$ and $CHBr_3$ as well as CFC-113 and CFC-11 were undetectable in subsequent blank gas measurements, residues of many hydrocarbons were persistent, suggesting a system contamination. In summary, the HayeSep D trap showed an overall higher number of residues which is likely caused by a higher desorption temperature of the Unibeads 1S trap which can be heated faster and to a higher temperature without degrading the material. Nevertheless, the residues on both adsorptive materials were on a tolerable level (below average measurement precision) for flask measurements with multiple measurements per sample.

**4 Application**

**4.1 Laboratory operation: flask sample measurements**

To ensure internal consistency of our laboratory instrumentation, five air samples were analysed with the GC-TOFMS instrument (Obersteiner et al., 2016) 
[revised manuscript text omitted]
$. Depending on GC- and detection system, this could induce artefacts in the detected signals and  also due to this limitation, the current configuration is not applicable to highly volatile compounds like $CF_4$, $C_2F_6$ or $C_2H_6$. Cooling capacity should however be sufficient to ensure quantitative trapping of such compounds on a suitable adsorptive material. Therefore, a starting point for future improvement is removal of $CO_2$ to extend the already large substance range by compounds of higher volatility. Regarding desorption, no blank residues were found for halocarbons that would cause concern or render the setup unsuited for halocarbon analysis (see "Appendix B: Blank Residues"). Relatively large amounts of hydrocarbons remained in blank measurements. These  residues are not an inherent problem of the preconcentration setup but more likely due to the adsorptive materials, carrier gas or valve membrane materials used. We do not attempt to present a viable correction method for any of the encountered analyte residues here. More dedicated experiments are necessary to account for analyte-specific memory effect and/or contaminations e.g. by modelling the carry-over from one sample to another and subtracting contamination background. By doing so, the applicability of the preconcentration unit can potentially be extended to quantitative hydrocarbon analysis.

[revised manuscript text omitted]
 ozonedeleting substances, their replacements, and related species. Ko, M.K.W., Newman P.A., Reimann, S. and Strahan, S.E.

(Eds.), SPARC Report No. 6, WCRP-15/2013, 2013. Available at www.sparc-climate.org/publications/sparc-reports/.

Velders, G. J., Fahey, D. W., Daniel, J. S., McFarland, M., and Andersen, S. O.: The large contribution of projected HFC emissions to future climate forcing, Proc. Natl. Acad. Sci. U. S.

A., 106, 10949-10954, doi: 10.1073/pnas.0902817106, 2009.

**1 Tables**

**Table 1.** (NEW) Technical configuration of the three preconcentration setups we operate. For futher
details on the full instruments (e.g. gas chromatography or detection), please refer to the respective
references. $T_D$ is given as a temperature range as it can be determined only indirectly (see sect. 2.3).

| Instrument | GhOST-MS (in-situ) | GC-QPMS (laboratory) | GC-TOFMS (laboratory) |
|---|---|---|---|
| Reference | Sala et al. 2014 | Hoker et al. 2015 | Obersteiner et al. 2016 |
| Adsorptive Material, type | HayeSep D, VICI, Switzerland | HayeSep D | HayeSep D (default) / Unibeads 1S, Grace, USA (testing purposes) |
| Adsorptive Material, approx. packed volume [mm$^3$] | 12 | 20 | 20 |
| Stirling Cooler | SC-TD08, Twinbird, Japan | M150, Global Cooling, USA (not available anymore) | CryoTel CT, Sunpower (Ametek), USA |
| $T_A$ [°C], routine operation | <−70, depending on ambient temperature as cooler operates at limit | −80 (cooling capacity would allow −120) | −80 (cooling capacity would allow <−120) |
| $T_D$ [°C] | 180-220 | 180-220 | 180-220 |
| reference volume [L] | 2 (1 tank) | 2-16 (4 tanks) | 2 (1 tank) |
| pressure sensor | Setra 204E, Setra Systems, USA | Setra 204, Setra Systems | Baratron 626, MKS Instruments, Germany |
| MFC | IQ-Flow IQF-200C, Bronkhorst, the Netherlands | EL-FLOW F-201CM, Bronkhorst | EL-FLOW F-201CM |
| Evacuation Pump | MD-1 vario SP, Vacuubrand, Germany | Trivac NT 5, Leybold (Oerlikon), Germany | MD-1 vario SP |
| Control/Operation | LabVIEW & cRIO, National Instruments, USA | LabVIEW & cRIO | LabVIEW & cRIO |

**Table 2.** Cycle times at $T_A$ of -80 °C / -120 °C (laboratory setup) and -70 °C (in-situ setup), based on
operational data. Laboratory setup configuration: Sunpower CryoTel CT Stirling cooler, preconcentra-
tion volume: 500 mL at 100 mL·min$^{-1}$, $T_D \approx 200$ °C for 3 min. In-situ setup configuration: Twinbird
SC-TD08 Stirling cooler, preconcentration volume: 100 mL at 100 mL·min$^{-1}$, $T_D \approx 200$ °C for 1 min.
Adsorptive material, both setups: HayeSep D. Due to a smaller coldhead, cooling rate and warm-up
during desorption are considerably larger with the in-situ setup, despite the shorter desorption time.

| $T_A$ [°C] | cooling rate at $T_A$ [°C·min$^{-1}$] | warm-up during desorption [°C] | minimum preconcentration cycle time  [min] | (NEW) Experimental total time needed for one measurement [min] |
|---|---|---|---|---|
| Laboratory instrument (GC-TOFMS) | | | | |
| −80 | −2.2 | 7.7 | 8.5 | 19.6 |
| −120 | −1.2 | 16.3 | 18.6 | 19.6 |
| In-situ instrument (GhOST-MS) | | | | |
| −70 | −4.1 | 13.5 | 4.1 | 4.1 |

**Table 3**. Results from a volume variation experiment, comprising measurements of the same reference
air with preconcentration volumes (PrcVol) of up to 2, 5 and 10 L. Laboratory setup, adsorptive mate-
rial Unibeads 1S. Volume-corrected detector response is referenced against calibration preconcentra-
tion volumes of 1 L (rR). rR <100% indicates underestimation (e.g. loss by breakthrough); rR >100%
indicates overestimation (e.g. increase by a memory effect from the preceding sample or contamina-
tion). Breakthrough is observed for COS at a preconcentration volume of 10 L while ethyne shows
signs of a system contamination (rR >100% despite a higher volatility compared to COS). CFC-12 and
CFC-11 show no indication of breakthrough, with all deviations from 100% rR below 3 σ.

| Substance | PrcVol [L] | rR | rR: 1 σ | PrcVol [L] | rR | rR: 1 σ | PrcVol [L] | rR | rR: 1 σ |
|---|---|---|---|---|---|---|---|---|---|
| Ethyne (C$_2$H$_2$) | | 102.0% | 0.66% | | 108.9% | 0.70% | | 109.2% | 0.70% |
| Carbonyl sulfide (COS) | 2 | 102.2% | 0.82% | 5 | 100.9% | 0.81% | 10 | 64.8% | 0.52% |
| CFC-12 (CCl$_2$F$_2$) | | 99.9% | 0.41% | | 100.7% | 0.42% | | 100.6% | 0.42% |
| CFC-11 (CCl$_3$F) | | 100.2% | 0.21% | | 100.5% | 0.22% | | 100.6% | 0.22% |

**Table 4. (Updated Values)** Retention times $t_R$ with relative standard deviations rSD and variances   for selected substances (same as **Table 3**) as well as their respective average signal width expressed as FWHM . Values derived  as arithmetic means over 4 measurement series from different dates (April 2015 to June 2016), comprising 149 individual measurements (~37 per series) of 19 different ambient air samples using the ramped GC program. For retention time variance, maximum to minimum differences over the 4 measurement series are given in brackets. Trap adsorptive material: HayeSep D. HFC-23 is the first detectable substance, least separated by chromatography. CFC-11 can be considered a reference for optimal chromatographic performance of the given setup.

[revised manuscript text omitted]

**Supplementary Information**

**Table 5** shows a list of substances detected up to the time of completion of  this paper. Identifications based on ambient air samples as well as synthetic mixtures. Substances are separated into six classes (e.g. CFCs and HCFCs, PFCs and HFCs etc.), which are listed in arbitrary order. Within each class, substances are sorted according to their boiling point (bp) in [°C]. Chemical sum formula as well as retention time $t_R$ in [min] on the GS GasPro PLOT column listed in columns two and three. Columns 5 & 6 show analyte residues in [%], expressed as chromatographic signal area determined in a blank gas measurement relative to a signal area determined in a preceding 1 L ambient air sample. Blank gas: purified helium 6.0 (Praxair, Germany). "Residue HayeSep D" denotes residues found with HayeSep D as adsorptive material, "Residue Unibeads 1S" shows the same for Unibeads 1S as adsorptive material. Residues that a constant background (contamination), are marked with a "*c*", ones that represent a memory effect from a preceding sample are marked with an "*m*". Substances that are not detected regularly in ambient air samples or show poor measurement precision $\geq 10~\%$ were excluded from the analysis ("not analysed"; n.a.). If no residue was detected or the detected residue was $\leq 0.01~\%$, a "not detected" (n.d.) is assigned to the respective substance.

**Table 5.** List of detectable substances and blank residues. Descriptions are given in the text.

| Class/Name | Formula | $t_R$ [min] | bp [°C] | Residue HayeSep D | Residue Unibeads 1S |
|---|---|---|---|---|---|
| ***CFCs & HCFCs*** | | | | | |
| HCFC-22 | $CHClF_2$ | 5.20 | -41 | n.d. | n.d. |
| CFC-115 | $CClF_2CF_3$ | 4.48 | -39 | n.d. | n.d. |
| CFC-12 | $CF_2Cl_2$ | 5.02 | -30 | n.d. | n.d. |
| HCFC-124 | $CHF_2CF_2Cl$ | 6.85 | -12 | n.d. | n.d. |
| HCFC-142b | $CH_3CClF_2$ | 6.87 | -10 | n.d. | n.d. |
| HCFC-31 | $CH_2ClF$ | 6.40 | -9 | n.a. | n.a. |
| CFC-114 | $CClF_2CClF_2$ | 6.67 | 4 | n.d. | n.d. |
| HCFC-133a | $C_2H_2ClF_3$ | 7.55 | 6 | n.d. | n.d. |
| HCFC-21 | $CHFCl_2$ | 7.32 | 9 | n.d. | n.d. |
| CFC-11 | $CFCl_3$ | 7.28 | 24 | n.d. | n.d. |
| HCFC-141b | $CH_3CCl_2F$ | 8.42 | 32 | n.d. | n.d. |
| HCFC-1121 | $CHClCFCl$ | 8.05 | 35 | n.a. | n.a. |

| Class/Name | Formula | $t_R$ [min] | bp [°C] | Residue HayeSep D | Residue Unibeads 1S |
|---|---|---|---|---|---|
| HCFC-132b | $CH_2ClCClF_2$ | 9.08 | 46 | n.d. | n.d. |
| CFC-113 | $CCl_2FCClF_2$ | 8.45 | 48 | 0.2% (m) | n.d. |
| HCFC-225ca | $CF_3CF_2CHCl_2$ | 9.37 | 51 | n.a. | n.a. |
| HCFC-225cb | $CClF_2CF_2CHClF$ | 9.57 | 56 | n.a. | n.a. |
| CFC-112 | $CFCl_2CFCl_2$ | 10.33 | 92 | n.d. | n.d. |
| HCFC-131 | $CCl_3CH_2F$ | 12.38 | 103 | n.a. | n.a. |
| | | | | | |
| ***PFCs & HFCs*** | | | | | |
| HFC-23 | $CHF_3$ | 3.01 | -82 | 2.6% (c) | n.a. |
| HFC-41 | $CH_3F$ | 4.38 | -78 | n.a. | n.a. |
| HFC-32 | $CH_2F_2$ | 4.20 | -52 | n.d. | n.d. |
| HFC-125 | $CHF_2CF_3$ | 4.87 | -49 | 0.4% (c) | 1.3% (c) |
| HFC-143a | $CH_3CF_3$ | 5.00 | -48 | n.d. | n.d. |
| HFC-161 | $C_2H_5F$ | 6.85 | -38 | n.a. | n.a. |
| PFC-218 | $C_3F_8$ | 4.02 | -37 | n.d. | n.d. |
| PFC-216 | $C_3F_6$ | 4.58 | -30 | n.a. | n.a. |
| HFO-1234yf | $CHFCHCF_3$ | 5.72 | -28 | 6.9% (c) | 14.9% (c) |
| HFC-134a | $CH_2FCF_3$ | 5.92 | -26 | n.d. | n.d. |
| HFC-152a | $CH_3CHF_2$ | 6.53 | -25 | n.d. | n.d. |
| HFC-134 | $CHF_2CHF_2$ | 6.32 | -23 | 1.1% (c) | 3.0% (c) |
| HFC-227ea | $CF_3CHFCF_3$ | 6.52 | -16 | n.d. | n.d. |
| HFO-1234ze | $CHFCHCF_3$ | 6.27 | -16 | n.d. | n.d. |
| PFC-318 | $c\text{-}C_4F_8$ | 5.68 | -6 | n.d. | n.d. |
| HFC-236fa | $CF_3CH_2CF_3$ | 7.22 | -1 | n.d. | n.d. |
| HFC-329ccb | $C_4HF_9$ | 7.67 | 15 | n.a. | n.a. |
| HFC-245fa | $CF_3CH_2CHF_2$ | 7.92 | 15 | n.d. | n.d. |
| HFO-1233zd | $CHClCHCF_3$ | 7.82 | 19 | n.a. | n.a. |
| HFC-356mff | $C_4H_4F_6$ | 8.35 | 25 | n.a. | n.a. |
| HFC-365mfc | $CF_3CH_2CF_2CH_3$ | 9.27 | 40 | n.a. | n.a. |
| | | | | | |
| ***Halons*** | | | | | |
| Halon-1301 | $CBrF_3$ | 3.87 | -58 | n.d. | n.d. |
| Halon-1211 | $CBrClF_2$ | 6.32 | -4 | n.d. | n.d. |
| Halon-1202 | $CF_2Br_2$ | 7.45 | 23 | n.a. | n.a. |
| Halon-2402 | $CBrF_2CBrF_2$ | 8.53 | 47 | n.d. | n.d. |
| Halon-2311 | $CF_3CHBrCl$ | 9.30 | 50 | n.a. | n.a. |

| Class/Name | Formula | $t_R$ [min] | bp [°C] | Residue HayeSep D | Residue Unibeads 1S |
|---|---|---|---|---|---|
| **_Chloro-, Bromo- & Iodocarbons_** | | | | | |
| Chloromethane | $CH_3Cl$ | 6.02 | -24 | 0.5% (c) | 0.6% (c) |
| Bromomethane | $CH_3Br$ | 7.00 | 4 | 3.4% (c) | 1.8% (c) |
| Chloroethane | $C_2H_5Cl$ | 7.92 | 12 | 25.5% (c) | 8.6% (c) |
| Dichloromethane | $CH_2Cl_2$ | 8.17 | 40 | 0.4% (c, m) | 0.2% (c) |
| Iodomethane | $CH_3I$ | 8.00 | 42 | 43.9% (c, m) | 46.2% (c, m) |
| Trichloromethane | $CHCl_3$ | 8.92 | 61 | 1.4% (c, m) | 0.7% (c, m) |
| Bromochloromethane | $CH_2BrCl$ | 9.03 | 68 | n.d. | n.d. |
| Methyl chloroform | $CH_3CCl_3$ | 9.93 | 74 | n.d. | n.d. |
| Tetrachloromethane | $CCl_4$ | 9.08 | 77 | 1.1% (m) | n.d. |
| Trichloroethene | $C_2HCl_3$ | 9.55 | 87 | n.d. | n.d. |
| Bromodichloromethane | $CHBrCl_2$ | 10.10 | 90 | n.d. | n.d. |
| Dibromomethane | $CH_2Br_2$ | 10.03 | 96 | n.d. | n.d. |
| Dibromochloromethane | $CHBr_2Cl$ | 11.53 | 119 | n.d. | n.d. |
| Tetrachloroethene | $C_2Cl_4$ | 10.62 | 121 | 23.9% (c, m) | 5.2% (c, m) |
| Tribromomethane | $CHBr_3$ | 13.50 | 147 | 11.2% (m) | n.d. |
| Diiodomethane | $CH_2I_2$ | 15.00 | 181 | n.a. | n.a. |
| **_Sulfur-containing and other halogenated compounds_** | | | | | |
| Sulfuryldifluoride | $SO_2F_2$ | 4.20 | -55 | n.d. | n.d. |
| Carbonyl sulfide | COS | 3.77 | -50 | 0.4% (c) | 0.1% (c) |
| Chlorotrifluoroethylene | $C_2F_3Cl$ | 4.92 | -28 | n.a. | n.a. |
| Perfluorotetrahydrofuran | $C_4F_8O$ | 5.87 | 2 | n.a. | n.a. |
| 3-chloropentafluoropropene | $CF_2CFCF_2Cl$ | 8.07 | 8 | n.d. | 7.6% (c) |
| Desflurane | $CF_3CHFOCHF_2$ | 8.42 | 24 | n.a. | n.a. |
| Carbon disulfide | $CS_2$ | 6.54 | 46 | 4.0% (c) | 0.8% (c) |
| Isoflurane | $CHF_2OCHClCF_3$ | 9.83 | 49 | n.a. | n.a. |
| Sevoflurane | $CF_3CF_3CHOCH_2F$ | 10.35 | 59 | n.a. | n.a. |

| Class/Name | Formula | $t_R$ [min] | bp [°C] | Residue HayeSep D | Residue Unibeads 1S |
|---|---|---|---|---|---|
| ***Hydrocarbons and Aldehydes*** | | | | | |
| Ethyne | $C_2H_2$ | 3.75 | -81 | 0.3% (c) | 1.4% (c) |
| Propene | $C_3H_6$ | 5.38 | -48 | 35.2% (c) | 28.5% (c) |
| Propane | $C_3H_8$ | 4.09 | -42 | 0.4% (c) | 0.1% (c) |
| Propyne | $C_3H_4$ | 7.17 | -23 | n.d. | n.d. |
| Formaldehyde | $CH_2O$ | 7.62 | -19 | n.a. | n.a. |
| Isobutane | $C_4H_{10}$ | 5.79 | -13 | 0.7% (c) | 1.0% (c) |
| Isobutene | $C_4H_8$ | 7.32 | -7 | n.d. | 75.3% (c) |
| 1-butene | $C_4H_8$ | 7.38 | -6 | n.a. | n.a. |
| 1,3-butadiene | $C_4H_6$ | 7.32 | -4 | n.a. | n.a. |
| n-butane | $C_4H_{10}$ | 6.05 | -1 | 0.3% (c) | 0.1% (c) |
| trans-2-butene | $C_4H_8$ | 7.02 | 1 | 25.3% (c) | 19.8% (c) |
| cis-2-butene | $C_4H_8$ | 7.24 | 4 | n.a. | n.a. |
| Acetaldehyde | $C_2H_4O$ | 11.26 | 20 | 99.2% (c, m) | 82.0% (c, m) |
| 2-methylbutane | $C_5H_{10}$ | 7.40 | 28 | 0.4% (m) | 0.2% (m) |
| Isoprene | $C_5H_8$ | 8.67 | 34 | n.a. | n.a. |
| n-pentane | $C_5H_{12}$ | 7.57 | 36 | 0.7% (m) | 0.3% (m) |
| trans-2-pentene | $C_5H_{10}$ | 8.47 | 36 | n.d. | 22.2% (c, m) |
| cis-2-pentene | $C_5H_{10}$ | 8.56 | 37 | n.a. | n.a. |
| 2-methylpentane | $C_6H_{14}$ | 8.61 | 60 | 0.8% (m) | 1.0% (m) |
| 3-methylpentane | $C_6H_{14}$ | 8.71 | 63 | 1.8% (m) | n.d. |
| n-hexane | $C_6H_{14}$ | 8.71 | 68 | 1.5% (c) | n.d. |
| Benzene | $C_6H_6$ | 11.00 | 80 | 2.5% (c) | 5.2% (c) |
| Cyclohexane | $c-C_6H_{12}$ | 8.82 | 81 | n.d. | n.d. |
| n-heptane | $C_7H_{16}$ | 10.06 | 98 | 23.1% (c, m) | 4.0% (m) |
| Toluene | $C_7H_8$ | 14.52 | 111 | 17.4% (c, m) | 9.8% (c, m) |